# Malleable, printable, bondable, and highly conductive MXene/liquid metal plasticine with improved wettability

Haojie Jiang[1], Bin Yuan[1], Hongtao Guo[1], Fei Pan[1], Fanmao Meng[1], Yongpeng Wu[2], Xiao Wang[1], Lingyang Ruan[1], Shuhuai Zheng[1], Yang Yang[1], Zheng Xiu[1], Lixin Li[1], Changsheng Wu [3,4,5,6], Yongqing Gong[1], Menghao Yang [1] & Wei Lu [1] ✉

Integration of functional fillers into liquid metals (LM) induces rheology modification, enabling the free-form shaping of LM at the micrometer scale. However, integrating non-chemically modified low-dimensional materials with LM to form stable and uniform dispersions remain a great challenge. Herein, we propose a solvent-assisted dispersion (SAD) method that utilizes the fragmentation and reintegration of LM in volatile solvents to engulf and disperse fillers. This method successfully integrates MXene uniformly into LM, achieving better internal connectivity than the conventional dry powder mixing (DPM) method. Consequently, the MXene/LM (MLM) coating exhibits high electromagnetic interference (EMI) shielding performance (105 dB at 20 µm, which is 1.6 times that of coatings prepared by DPM). Moreover, the rheological characteristic of MLM render it malleable and facilitates direct printing and adaptation to diverse structures. This study offers a convenient method for assembling LM with low-dimensional materials, paving the way for the development of multifunctional soft devices.

Gallium-based liquid metals (LM)[1–5], which combine the fluidity of liquids with the electrical conductivity of metals, have found widespread applications in fields such as flexible electronics[4,6–9], electromagnetic shielding[10–12], thermal interface materials[13,14], and soft robotics[15,16]. The formability and patterning capabilities of LM determine its electrical behavior and utility and are integral to functional device manufacturing. However, LM exhibits significant interfacial tension, making the conversion into non-spherical, high surface area shapes challenging.

Existing methods for preparing LM-based conductive functional materials typically involve interface engineering of LM droplets, followed by blending with reinforcing phases[17–22]. Additionally, LM composites are generally added to insulating substrates[23–25] to achieve specific shapes and mechanical strength. However, the challenges of the decreased conductivity and compromised mechanical performance of the substrate remain unresolved. Furthermore, the limited wettability of LM at non-reactive interfaces[6,26] and the secondary sintering[17,27–30] required for conductive path activation pose challenges in shaping them on flexible substrates or complex surfaces. Although LM conduction paths can be constructed directly in various matrices[10,16,31], it increases the complexity and technical difficulty of the preparation process. To address these issues, rheological modification was applied to LM to prepare composite conductors with LM as a continuous phase, improving wettability and enabling better shape control and higher precision manufacturing.

[1]Shanghai Key Lab. of D&A for Metal Functional Materials, School of Materials Science & Engineering, Tongji University, Shanghai 201804, China. [2]School of Materials and Chemistry, University of Shanghai for Science and Technology, Shanghai 200093, China. [3]Department of Materials Science and Engineering, National University of Singapore, Singapore 117575, Singapore. [4]Department of Electrical and Computer Engineering, National University of Singapore, Singapore 117583, Singapore. [5]Institute for Health Innovation and Technology, National University of Singapore, Singapore 117599, Singapore. [6]The N.1 Institute for Health, National University of Singapore, Singapore 117456, Singapore. ✉e-mail: weilu@tongji.edu.cn

Utilizing liquid-phase LM as a dispersing solvent, metal particles such as Cu[32,33], Ag[34], and Ni[35] are mechanically drawn into the LM, gradually forming metal intermetallic compounds to increase the roughness of the LM and enhance its adhesion to the substrate. However, the high density of the metal itself and the continuous mutual reaction between LM and the metal particles are detrimental to the stability of the materials. Furthermore, nonmetallic low-dimensional materials with specific properties, such as carbon nanotubes (CNTs)[36], graphene oxide (GO)[37], and MXene[38], are typically challenging to mix with LM through simple mechanical stirring and often require precise size control and chemical modification, which restricts the manifestation of the inherent characteristics of the incorporated medium and the functionalization of the LM. In addition, the aggregation of particles and subsequent macroscopic phase segregation have hindered the development of LM-second phase composite materials, primarily due to the unique liquid environment of LM. Hence, it is worthwhile to develop effective approaches for rapidly fabricating LM composites without intermetallic compounds or third-phase media.

Herein, we propose a solvent-assisted dispersion (SAD) method that utilizes fluid and liquid metal deformation characteristics to fabricate malleable MLM composites. SAD utilizes the fragmentation-reintegration of LM in non-toxic volatile solvents to achieve mutual-dispersion with MXene without the need for additional residual media, significantly enhancing the uniformity and stability of filler dispersion compared to traditional dry powder mixing (DPM). SAD is also applicable to mix other fillers such as carbon nanotubes, graphene, and magnetic particles with LM. The exceptional deformability of the prepared MLM, coupled with robust adhesion to various flexible substrates, makes it highly versatile for complex surface applications and sensing capabilities. The MLM coating prepared by the doctor-blade method exhibited an EMI shielding performance of 105 dB at a thickness of 20 μm and a high shielding efficiency. Moreover, the high Joule heating performance of thin MLM coatings expands their potential for practical applications, such as thermal therapy or surface deicing. Additionally, the programmed electronic paste printing of MLM also demonstrates its potential in integrated circuit manufacturing. Therefore, the MLM material with exceptional scalability holds broad prospects in electromagnetic protection, flexible electronics, and advanced manufacturing.

## Results

### Design and characterization of MLM

The incorporation of MXene into LM is challenging. The preparation process and mechanism of compounding MXene powder with LM to form MXene/LM composite by DPM (MLM-D) is displayed in Supplementary Fig. 1. Unlike most metal powders that form intermetallic compounds after blending with LM[33,35,39], which decreases the fluidity of LM, the MLM-D system only relies on continuous mechanical deformation to manipulate the oxide layer on the surface of LM to change the rheological properties of the mixture. However, the absence of a hydrogen bond environment in LM prevents MXene from forming a stable dispersion. Only stirring can be relied upon to minimize the aggregation of MXene powder in high-viscosity LM. However, DPM ultimately fails to guarantee the uniform dispersion of MXene within the LM. In particular, the size-dependent effect in conventional particle doping requires specific particle dimensions to form stable plasticine-like LM complexes[37]. SAD, on the other hand, can effectively address this problem, as it can infiltrate particles across a wide range of scales, from nanometers to micrometers. Herein, we primarily focused on the formation of the MXene/LM composite by SAD (MLM-S), as illustrated in Fig. 1a. This methodology can be applied to form other LM composites with metallic or nonmetallic solid particle fillers to meet specific requirements. We demonstrated its effectiveness with three successful cases of particle infiltration: difficult-to-disperse

multi-walled carbon nanotubes, graphene, and nano $Fe_3O_4$ particles (Supplementary Table 1).

In a typical MLM-S preparation process through SAD (Fig. 1a), $Ti_3AlC_2$ MAX was first selectively etched with HCl/LiF. Then, the MXene paste obtained by centrifugal replacement of the water in the MXene aqueous dispersion with anhydrous ethanol (EtOH) was transferred to a small PTFE beaker for magnetic stirring (Supplementary Fig. 2). Subsequently, an appropriate amount of LM was gradually added dropwise with a plastic syringe. Next, the mixture in the uncapped beaker was continuously stirred until EtOH had completely evaporated. EtOH and minute traces of residual water serve to create a hydrogen-bonding environment for the dispersion of MXene while concurrently reducing the yield stress of the LM and facilitating the formation of a slip layer between LM and MXene[2]. The mechanical stirring process leads to the fragmentation of LM into tiny droplets. The solvent slip layer reduces the interfacial tension between LM and its surroundings, preventing LM droplets from re-agglomerating. Thus, the LM is gradually dispersed and encapsulated within the MXene paste. As EtOH evaporates, the volume of the MXene paste decreases, and the fragmented and refined LM begins to gradually aggregate and come into contact, which then engulfs the original MXene paste. During the stirring process, MXene is uniformly dispersed, and when EtOH completely evaporates, the reintegrated LM matrix fully encapsulates the MXene. The optical images and contact angles at each stage, presented in Supplementary Fig. 3, illustrate that the introduction of MXene into LM by SAD improves the non-wetting behavior of LM on PTFE plates and enables complete spreading, demonstrating the adequate wetting of MLM-S on low-surface-energy surfaces. Another important aspect is that during the conventional DPM process, LM always remains a monolithic unit that encapsulates the MXene powder. Conversely, the SAD method involves mutual encapsulation and fusion. To confirm the microscopic distribution of the two phases, we used X-ray microscopy (XRM) to perform a Micro-CT scan of MLM-S. The density difference between MXene and LM enables the visualization of the biphasic MLM structure through the correlated density distributions, as measured by x-ray absorption (Supplementary Movie 1 and 2). Figure 1b shows the distribution of the two phases within a cube with a side length of 722 μm. MXene (blue) adheres to each other in nearly spherical morphology, uniformly distributes within the LM (orange) matrix, and forms continuous conductive paths. It was determined that MXene occupies 34 to 42% of MLM-S by volume with a weight percentage of 2%. The density of MLM was determined to range from 2.6 to 4.9 g cm$^{-3}$, depending on the MXene loading (Supplementary Fig. 4), lower than that of the ones prepared via MLM-D.

Besides the ubiquitous van der Waals interaction present in all systems, the binding between MXene and LM is critical, as it determines the stability of MLM-S. The MLM composite was detected by X-ray diffraction (XRD) to investigate the phase composition (Supplementary Fig. 5). The result shows that no new phases formed during the formation of the MLM plasticine. X-ray photoelectron spectroscopy (XPS) was carried out to determine the chemical states of both the pristine materials and the prepared plasticine. In contrast to the pristine MXene, the MLM composite exhibits additional subpeaks of In and $In_2O_3$ in the Ti 2p spectrum due to the overlapping with the In $3d_{3/2}$ (Fig. 2a), and the same phenomenon is observed in the In 3d high-resolution spectrum of the MLM (Supplementary Fig. 6c). Moreover, the MLM composite exhibits a shift of the Ti 2p peaks towards lower binding energy. This can be primarily attributed to the synergetic effects of defect formation and interfacial interactions between LM and MXene[40,41]. The shift to lower binding energies suggests an increase in the electron density around the Ti atoms, attributed to the hydrogen-bonding interaction between the metal oxide layer (Supplementary Fig. 6) and the hydroxyl groups on the MXene surface[42], facilitating the charge transfer. The redshift and broadening of

hydroxyl peaks in the MLM infrared spectrum further confirm the presence of hydrogen bonds (Supplementary Fig. 7).

To further elucidate the bonding between MXene and LM, DFT simulations were conducted using VASP to investigate the interfacial interactions between hydroxyl-terminated MXene and LM with an oxide layer. Specifically, six oxygen atoms were introduced into an amorphous $Ga_{2.5}In$ alloy model to simulate the formation of an oxide layer. After a simulation period of 1.7 ps (Fig. 2b, c), hydrogen bonds were observed between two oxygen atoms within the metal oxide layer and the hydroxyl terminations of MXene, ensuring a strong interaction between the two phases. Notably, the system energy stabilized after 1.7 ps in a 2.2-ps dynamic equilibrium process (Supplementary Fig. 8), further confirming the stability of the interface between MXene and EGaIn.

Therefore, the homogeneous dispersion of MXene in LM and the stable combination of the two phases prevent MLM-S from phase segregation and phase precipitation. Moreover, MLM-S retains the self-healing properties of LM, exhibiting great processability and versatility. It can be effortlessly shaped into various forms and bonded to each other (Fig. 2d, Supplementary Movie 3). In addition, the integration of MXene with LM significantly enhances the adhesion and wettability of LM to diverse substrates. MLM-S demonstrated a superior grip on smooth glass bevels, setting it apart from the original LM (Supplementary Fig. 9a, b). As depicted in Fig. 2e, a thin layer of MLM-S

coating can be quickly applied to the underside of a model car with a complex structure. The original structural features remain clearly visible after coating, demonstrating the excellent adaptability of MLM-S to complex surfaces. Notably, the high surface tension of pure LM limits its microelectronic printability[43]. The introduction of MXene improves the rheological properties of the LM, playing a vital role in enabling direct printing. As shown in Supplementary Fig. 10, the viscosity of MLM-S gradually increases with MXene content (from 82174 Pa·s for $M_{0.25}$LM-S to 814672 Pa·s for $M_3$LM-S at a shear rate of $0.013\,s^{-1}$). The transformation from a thin liquid state to a plasticine-like state is characterized by a significant increase in viscosity and the absence of flow unless physically agitated, as illustrated by a sample in a tilted beaker (Supplementary Fig. 11, Supplementary Movie 4). Additionally, the viscosity of the samples decreases rapidly with the shear rate, exhibiting shear-thinning behavior, which facilitates continuous extrusion of MLM-S. The low-viscosity $M_{0.25}$LM-S can be used directly as an ink for rapid printing of high-fidelity 2D patterns without mixing it with other insulating substances[44] (Supplementary Movie 5). The printed pattern can adapt to substrate deformation, and no re-fusion into large droplets was observed (Fig. 2f, g). Notably, large-scale coating is also feasible. As shown in Fig. 2h, a large-area MLM-S coating (50 cm × 60 cm) was achieved. However, it is important to note that the viscosity of MLM-S is too high for blade coating if the weight of infiltrated MXene exceeds 3.0 wt% (Supplementary Fig. 12). MLM-S can

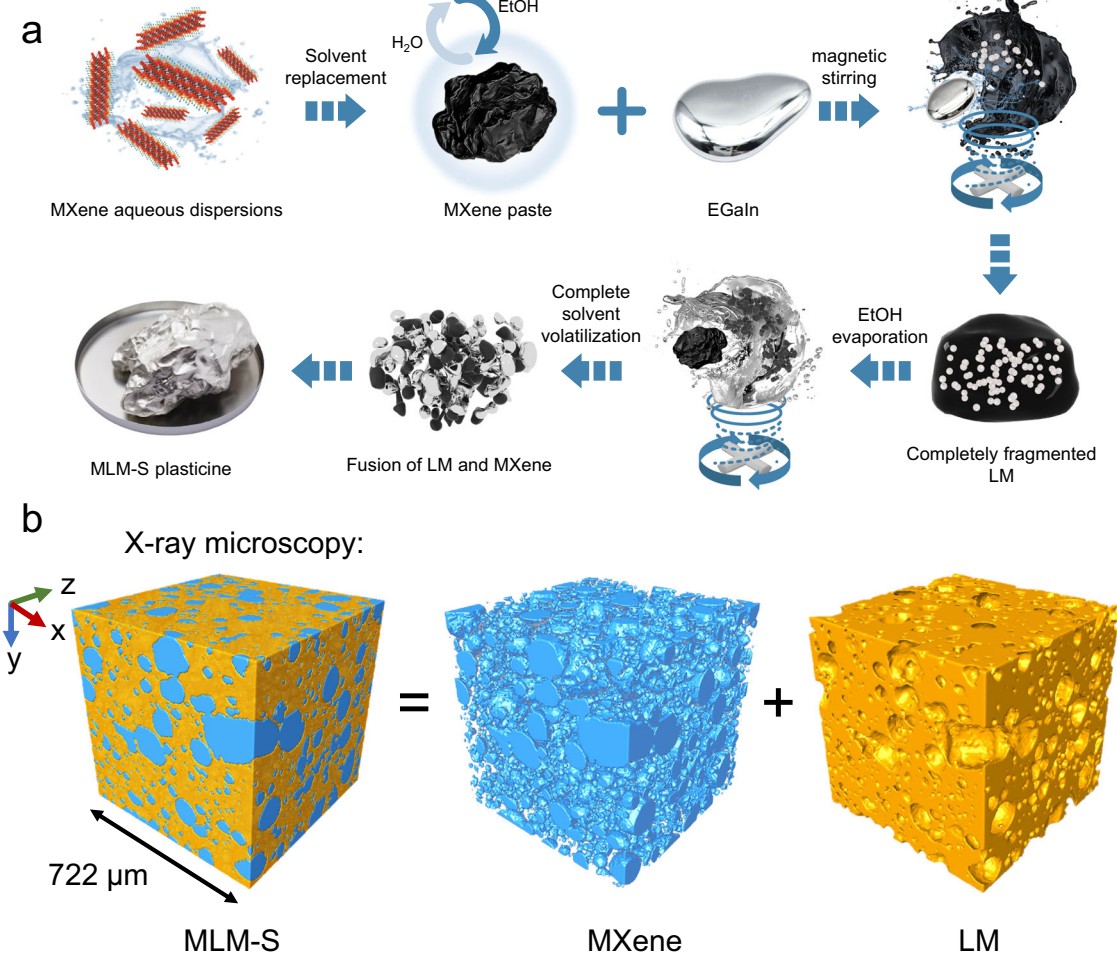

**Fig. 1 | Fabrication of the MXene/liquid metal composite by solvent-assisted dispersion (MLM-S). a** Schematic illustration of the preparation process of solvent-assisted dispersion (SAD). With the exception of the third image in the first row (EGaIn), which is a physical photograph, the rest of the images were generated using 3D modeling software. **b** X-ray microscopy (XRM) image demonstrating the bi-conductive phase structure of MLM-S. Two distinct domains—the MXene (blue) and liquid metal (LM, orange)—interpenetrate.

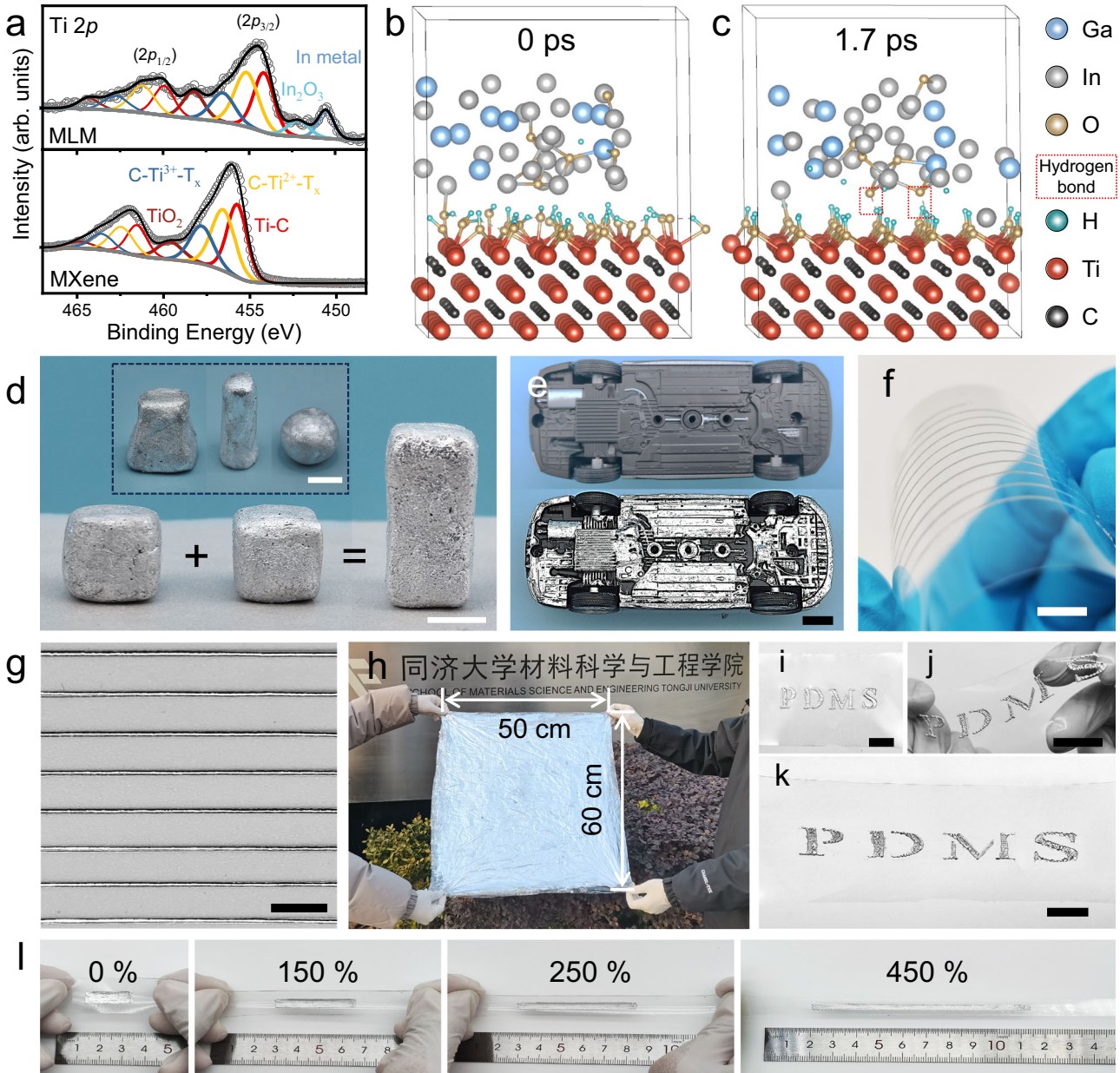

**Fig. 2 | Chemical characterization, formability, and patterning capabilities of the MXene/liquid metal composite prepared by solvent-assisted dispersion (MLM-S). a** Comparison of Ti 2*p* high-resolution XPS spectra of the MLM composites and MXene. Ab initio molecular dynamics (AIMD) Simulation of interface interaction between liquid metal (LM) and MXene at **b** 0 ps and **c** 1.7 ps. **d** MLM-S is highly processable and can be readily shaped into arbitrary shapes and adhere to each other. **e** Photos of the chassis of a model car before and after MLM-S coating. **f** Optical image of high-resolution MLM-S lines on a PET substrate fabricated through direct printing. The printed MLM-S can well adapt to deformation when the PET substrate is bent. **g** The magnified image of **f. h** Image of large-scale MLM-S coating (50 cm × 60 cm). **i** Application of MLM-S onto a polydimethylsiloxane (PDMS) film with a hollow letter template. MLM-S coated PDMS under twisting **j** and stretching **k. l** MLM-S coating under varying strains (from 0% to 450% from left to right). Scale bar, 1 cm.

be applied directly to a wide range of soft or rigid substrates, including paper, wood, plastics, fabrics (Supplementary Fig. 13a–d), and PDMS (Fig. 2i). It retained its fluidity to accommodate various substrate deformations, such as torsion (Fig. 2j) and tension (Fig. 2k). Furthermore, it exhibited exceptional resistance to delaminating even under substantial deformations (Fig. 2l, Supplementary Movie 6). Therefore, MLM-S enables rapid, efficient, and cost-effective patterning through precision printing with high design freedom or coating methods.

The microstructure of the MLM coating (~10 μm thick) on paper (Fig. 3a) was analyzed using scanning electron microscopy (SEM) and energy-dispersive X-ray spectroscopy (EDS). As presented in Supplementary Fig. 14e and the related EDS analysis (Supplementary Fig.

$14e_{1-4}$), prominent MXene agglomerates (marked by Ti elemental signal aggregation) and numerous defects (holes and cracks) can be observed on the surface of the MLM-D coating. Furthermore, the surfaces of MXene exhibits no Ga/In/O elemental signals (Supplementary Fig. $14e_{2-4}$, $f_{2-4}$), implying poor interaction between the two phases. In contrast, MXene is uniformly distributed in the MLM-S coating without any noticeable agglomeration (Fig. $3b_1$), and the MLM-S surface consists of a compact layer of Ga-In, as shown in Fig. $3_{2,3}$. Minute and minimal cracks and defects were observed on the surface of the coating, as depicted in the magnified image in Fig. 3c. However, it is noteworthy that these cracks do not penetrate through the coating. Figure $3c_1$ shows a slight exposure of MXene,

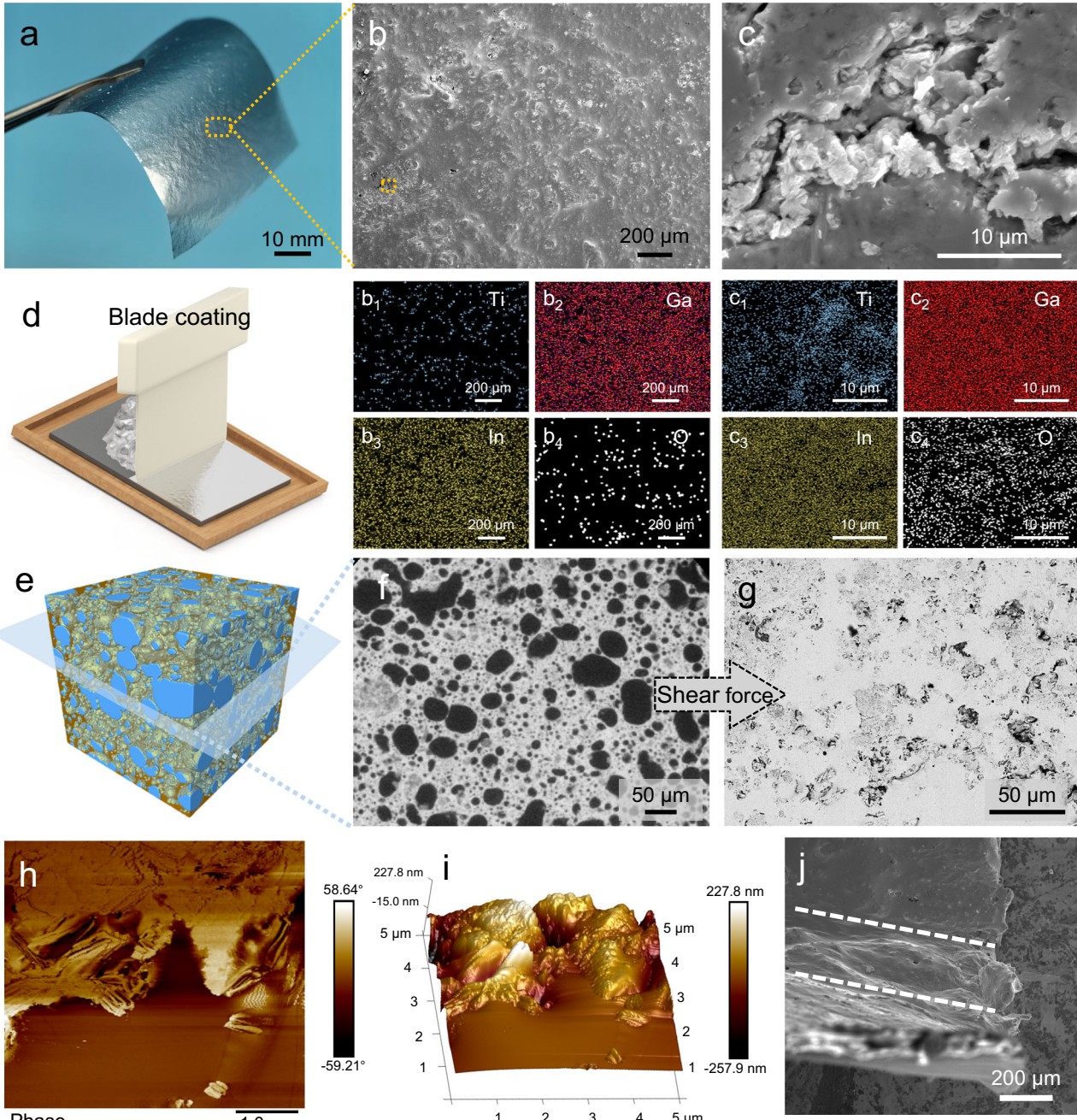

**Fig. 3 | Microscopic characterization of the MXene/liquid metal composite prepared by solvent-assisted dispersion (MLM-S). a** Photograph of an MLM-S coated weighing paper. **b** SEM image of the MLM-S coating surface. **b₁₋₄** EDS mapping of the region **b**. **c** SEM image enlarged from the orange box region in **b**. **c₁₋₄** EDS mapping of the region **c**. **d** Schematic diagram illustrating the blade coating of the MLM-S plasticine. The image is generated using 3D modeling software. **e** 3D reconstruction image of the MLM-S plasticine generated by collecting and reconstruction 2D projection images by X-ray microscopy (XRM). The blue part represents MXene, while the orange portion represents LM. The LM is made semi-transparent to more clearly show the MXene inside. **f** Slice image of **e** exhibiting the horizontal section. **g** Backscattered electron image of MLM-S coating surface. **h** AFM phase image and **i** AFM 3D height image of the M₂LM-S coating. **j** SEM image of the MLM-S coating under 90° bending. The area between the two white dashed lines is where the MLM-S coating bends.

while Ga-In and its oxide layer remain uniform on the surface (Fig. 3c₂₋₄), indicating the absence of phase separation. It could be attributed to the formation of hydrogen bonds between the LM oxide layer and MXene in the MLM-S, allowing both phases to deform synergistically under the shear force applied during coating. The 3D image (Fig. 3e) and its horizontal section (Fig. 3f) obtained by XRM further reveal the spatial distribution of MXene in the plasticine. The near-spherical MXene domains, uniformly dispersed in LM, is deformed into irregular morphology along with the LM on the substrate during coating, forming a continuous network (Fig. 3g), demonstrating the existence of a bicontinuous structure in the MLM-S layer. As seen in the AFM phase image (Fig. 3h), the variation in the vibrational phase of the probe, caused by the presence of different material phases, results in distinct colors in the phase diagram, showing the composition and distribution of the coating. The bright MXene region is wrapped by the dark metal oxide layer, while the brown LM almost covers nearly the entire scanning area. The AFM height image of the coating reveals the undulating surface

morphology of the MXene, contrasting with the comparatively smooth LM regions. (Fig. 3i).

Therefore, SAD optimizes the ability of LM to encapsulate MXene and improves the dispersion of MXene in the LM, facilitating the formation of a bicontinuous structure. Cross-sectional SEM images of the MLM-S coating show a conformal interface between the MLM-S and the substrate with no gaps or voids (Supplementary Fig. 15), revealing uniform Ti element distribution and Ga-In layer coverage(Supplementary Fig. 16). Supplementary Fig. 17 shows cross-sections of coatings with varying thicknesses. Moreover, the high adhesion and damage resistance of the coating under the substrate deformation are crucial for practical applications. As shown in the area between the dashed lines in Fig. 3j, the MLM-S coating adheres well to the substrate under severe deformation without cracking or delamination.

## EMI shielding performance and mechanism

The thickness of the shielding material is crucial for electromagnetic interference shielding effectiveness (EMI SE), especially with the miniaturization of electronic devices. LM is an excellent candidate for shielding electromagnetic waves (EMWs) due to its high electrical conductivity[35]. With the addition of MXene, the fluidity of LM decreases, and its adhesion to the substrate significantly improves. Therefore, MLM can serve as an effective EMI shielding coating. MLM coatings of varying thicknesses can be readily applied onto various substrates using a blade. EMI shielding materials of different thicknesses can be made by applying MLM-D and MLM-S with different compositions to ordinary weighing paper (Supplementary Movie 7). The comparison of the conductivity of different coatings in Fig. 4a reveals that the conductivity of LM remains virtually unaffected by the addition of MXene, which is beneficial for designing an ultra-thin EMI shielding material[45]. In contrast, the conductivity of MLM-D exhibits a significant decrease compared to that of the original LM. This can be attributed to the challenging interconnectivity between MXene powder particles, as well as the presence of large MXene agglomerates, which directly disrupt the conductive pathways of the LM (Supplementary Fig. 14f$_{2-3}$).

When using the plasticine prepared by DPM as an EMI shielding coating (thickness: 10 μm), the EMI SE of MLM-D is enhanced compared to MXene, LM, and Ag$_{10}$LM with the same thickness (Fig. 4b). As the MXene content increases, the EMI SE of the MLM-D increases until the MXene content exceeds 4 wt% (Fig. 4b). This could be attributed to the excessive introduction of MXene, which can lead to powder aggregation, resulting in an effect of $1 + 1 < 2$. As shown in Fig. 4c, the maximum average EMI SE in the X-band increases from 60.7 dB to 66.0 dB as the coating thickness of M$_2$LM-D is increased from 10 to 20 μm. Notably, Fig. 4c shows that the EMI SE of the MLM-D is similar to that of both pure LM and pure MXene film (thickness: 20 μm), implying that the unique two-dimensional structure and shielding advantages of MXene may not be efficiently utilized by DPM[46,47]. Supplementary Fig. 18 shows the EMI SE of MLM-S (MXene content: 2 wt%) coatings with thicknesses of 3, 5, 10, and 20 μm. In the whole X-band, the M$_2$LM-S coating can achieve a stable EMI shielding performance of about 85 dB at 10 μm and more than 100 dB at 20 μm, which are significantly higher than those of both pure LM and pure MXene films of the same thickness. The superior performance of MLM-S over MLM-D could be explained as follows: Compared to the MXene powder used in DPM, the MXene paste used in the SAD occupies a much larger volume for the same mass (Supplementary Fig. 19). In other words, the inter-flake spacing in the MXene paste is much larger, and the inter-flake interaction is much weaker, facilitating uniform dispersion in the LM in the subsequent SAD steps. This leads to a more favorable percolation state according to percolation theory.

To better understand the reasons behind the excellent EMWs attenuation achieved by the MLM-S coating, the total EMI shielding

efficiency ($SE_T$), microwave reflection ($SE_R$), and microwave absorption ($SE_A$) were calculated using Equations (3–6). As illustrated in Fig. 4d, the $SE_A$ of MLM-S coatings increases with the rise in coating thickness, while the $SE_R$ and $SE_T$ remain almost unchanged. Moreover, the dominant contribution to the overall EMI SE is from $SE_A$, accounting for 74.6–86.4% of the overall EMI SE, indicating the efficient attenuation of penetrated EMWs in the MLM-S coatings. In contrast, the $SE_A$ of MLM-D accounts for only 67.1–72.9% of the overall EMI SE. In addition, the average EMI SE of the M$_2$LM-S coating reaches 109.2 dB at a thickness of 20 μm, much higher than that of the M$_2$LM-D coating of the same thickness (61.6 dB). This substantial improvement is essential for developing efficient, lightweight shielding materials suitable for integration into wearable electronics applications. It should be note that addition of excess MXene (M$_3$LM-S) resulted in an 18.3% loss in EMI SE. Moreover, 20-μm-thick M$_2$LM-S coating demonstrates extraordinary EMI SE of ≈105 dB over a broadband frequency range (5.88–18 GHz) (Fig. 4e), highlighting its great performance.

The power coefficients of MLM-S coatings were calculated to elucidate the EMI shielding mechanism. As shown in Supplementary Fig. 20a, the reflection coefficient ($R$) of all coatings is much larger than that of the absorption coefficients ($A$), revealing that the primary shielding originates from reflection loss. The major EMI shielding mechanism in the MLM-S coating is schematically presented in Fig. 4f. The EMI shielding capability stems primarily from the exceptional conductivity of the MLM-S. The high conductivity results in numerous free charge carriers at the interface, causing severe impedance mismatch and therefore reflecting most of the incident EMWs. The residual EMWs penetrate the top surface of the coating and enter its interior. As the EMWs propagates through the LM/MXene bi-conductive conductive network, they interact with the free charge carriers, converting a substantial portion of the EWMs energy into thermal energy, thereby resulting in their attenuation. Moreover, the presence of free charges at the interfaces between the MXene sheets and the LM induces interfacial polarization, further contributing to the attenuation of EMWs. As a result, almost all of the penetrated EWMs are attenuated within the coating, causing the $T$ value to approach zero[17,45,48].

From a practical application perspective, reliable EMI shielding is critical. Benefiting from the deformability and cyclic stability of the LM matrix, the shielding performance of Ecoflex films coated with M$_2$LM-S remains stable even after 1000 cycles of 100% stretching (Supplementary Fig. 21). Moreover, Fig. 4g presents the EMI shielding performance of a 20-μm-thick M$_2$LM-S coating after being heat treated at 70 °C for 15 days, exhibiting almost no degradation compared to the untreated sample. Even when the plasticine is subjected to flame burning, coatings made of it still exhibit excellent EMI shielding performance, close to 90 dB (Fig. 4g). These results demonstrate that the reliable performance of MLM-S even at elevated temperatures and even under extreme conditions. The excellent resistance to high temperatures is attributed to the effective isolation and protection of the internal MXene from damage by the external LM. Additionally, we demonstrate the effective EMI shielding behavior of the MLM-S coating using a wireless power-transfer system, as shown in Supplementary Fig. 22.

The EMI $SSE_t$ (Fig. 4h) and $SE_T$ (Supplementary Fig. 23) of the M$_2$LM-S coatings are compared with previous studies (for more details refer to Supplementary Table 2). Metal-based and carbon-based EMI shielding materials have been prominent in recent years. However, the high density of metals restricts their suitability for certain advanced applications, and carbon-based materials with lower electrical conductivity often require increased thickness to achieve adequate EMI SE. While MXene films and MXene aerogels/foams exhibit excellent shielding performance, there is room for improvement in their mechanical strength, flexibility, and durability. Composite incorporating MXene with reinforcing phases can enhance overall strength, but

challenges such as the bucket effect need to be addressed. LM-based materials dispersed in reinforcing phases often experience electrical isolation due to the formation of oxide layers, leading to a loss in shielding capability and necessitating the pressure sintering process[17,30]. The MLM-S coating, featuring a biphasic conducting network, demonstrates a stable EMI SE of up to 105 dB and a specific EMI shielding effectiveness of up to 59336.8 dB cm² g⁻¹, showcasing its profound potential in EMI shielding and related applications.

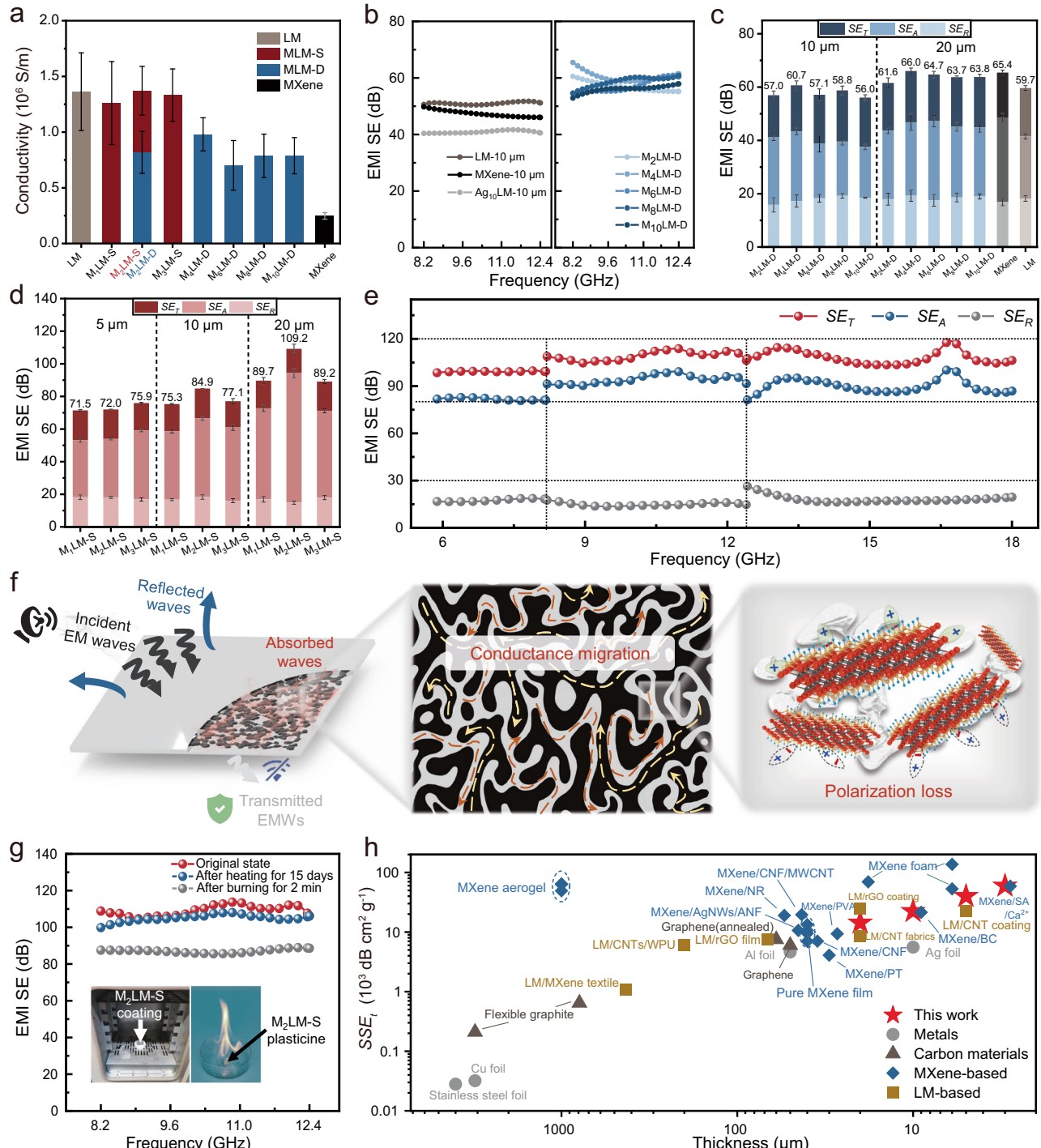

**Fig. 4 | Electrical properties and EMI SE of the MXene/liquid metal composite prepared by dry powder mixing (MLM-D) and solvent-assisted dispersion (MLM-S). a** Electrical conductivity of the MLM-D and MLM-S coatings with different MXene content. All values represent the mean ± standard deviation (SD) (n = 6). **b** EMI SE of the liquid metals (LM), MXene, the Ag nanosheets /liquid metal composite (Ag: 10 wt%) prepared by dry powder mixing (Ag₁₀LM), and MLM-D coatings with different MXene content. **c** EMI $SE_T$, $SE_A$, and $SE_R$ of the MLM-D coatings with thicknesses of 10 and 20 μm. Error bars represent the SD of 201 scanning points in the X-band. **d** EMI $SE_T$, $SE_A$, and $SE_R$ of MLM-S coatings with thicknesses of 5, 10, and 20 μm. Error bars represent the SD of 201 scanning points in the X-band. **e** EMI $SE_T$, $SE_A$, and $SE_R$ of the MLM-S coating with a thickness of 20 μm from 5.88 to 18 GHz. **f** Illustration of EMI shielding mechanisms for the MLM-S coating. The black and gray regions in the middle panel represent MXene and LM, respectively. These images are generated using 3D modeling software. **g** EMI SE of the MLM-S coatings before and after heat treatment. The insets show the heat treatment environment: a muffle furnace (at 70 °C for 15 days) (left) and the plasticine burning (for 2 min) (right). **h** Comparison of the $SSE_t$ of the M₂LM-S coating with that of other EMI shielding materials reported in the literature.

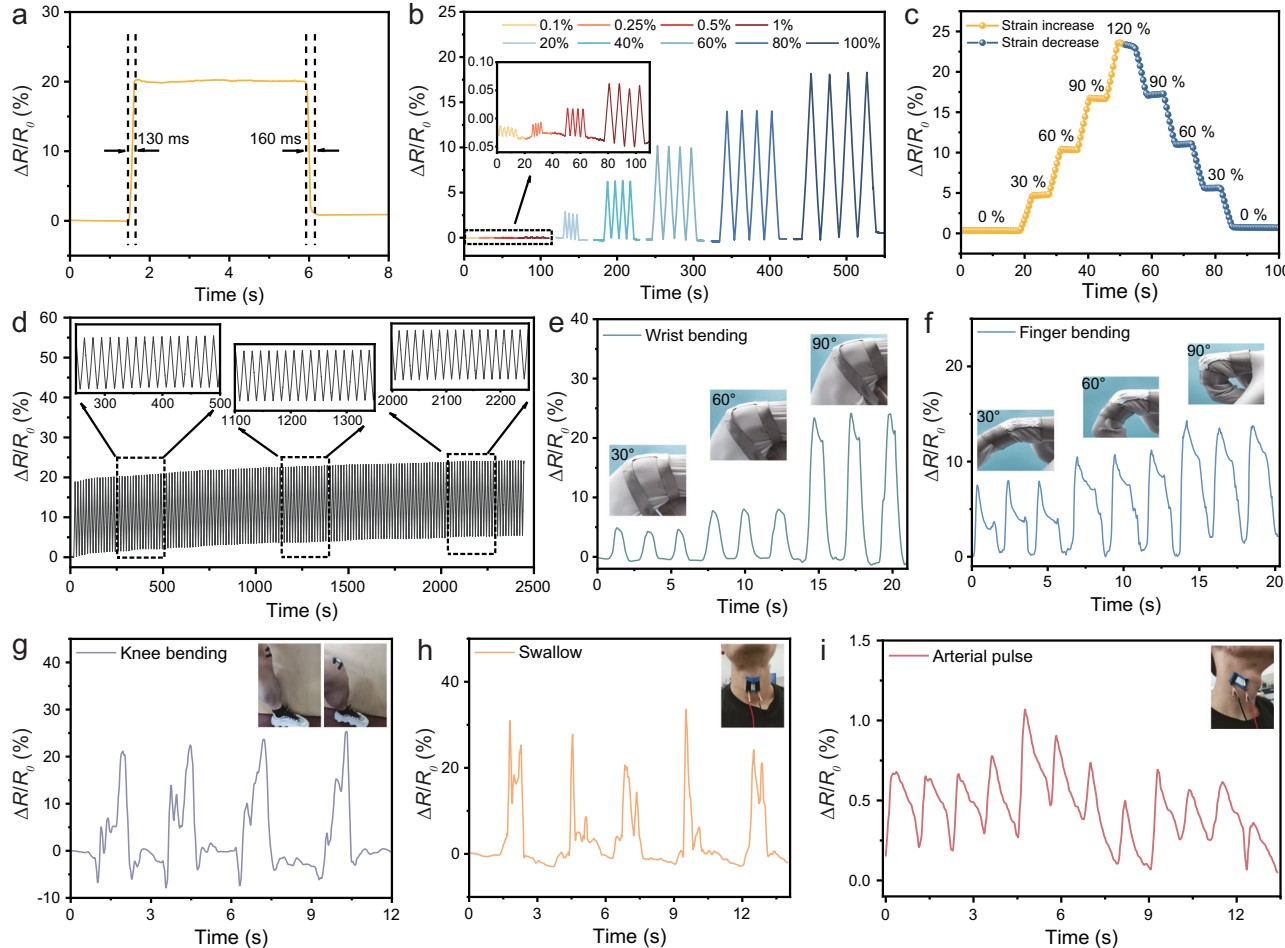

**Fig. 5 | Sensing properties and human motion detection of the MXene/liquid metal composite prepared by solvent-assisted dispersion (MLM-S). a** Sensing response and recovery time under 100% tensile strain. **b** Change of relative resistance ($\Delta R/R_O$) of the MLM-S sensor under 0.1%-100% tensile strain. **c** Change of relative resistance ($\Delta R/R_O$) in a strain cycle from 0% to 120% and then back to 0%, holding for 5 s at different strain levels. **d** Cyclic tensile test of the sensor. Sensing response to wrist **e** and finger **f** bending at different angles, including 30°, 60° and 90°. Sensing response to knee bending **g**, swallow **h**, and arterial pulse **i**.

## Sensing property and human motion detection

Thanks to its plasticity and improved wettability, MLM can be seamlessly coated and encapsulated in a flexible material, integrating high shielding with flexibility (tensile and flexural properties) to fulfill the shielding requirements of flexible electronics. In this study, Ecoflex, selected as a flexibility and stretchability, served as the substrate for both applying MLM-S as a coating and encapsulating it, enabling the recording of mechanical and complex muscle movements. As shown in Fig. 5a, the MLM-S sensor exhibits a very rapid response and recovery, with a response time of 130 ms and a recovery time of 160 ms. The fast response of the sensor is based on the ability of the coating to deform quickly and its strong adhesion to the substrate. Supplementary Fig. 24 illustrates the relationship between strain and the relative change in resistance ($\Delta R/R_O$). The sensitivity of the sensor is assessed by determining the gauge factor (GF), which is derived from the slope of the $\Delta R/R_O$ versus strain[49], indicating the GF of 0.14 in 0–200% strain. Figure 5b exhibits the variation of the resistance signal of MLM flexible films under different strains. The $R/R_O$ value increases dramatically when the applied strain increases and completely recovers to the initial value after release, exhibiting great stability and reversibility. Moreover, the flexible film can also produce a stable and recognizable signal for 0.1–1.0% strain. The flexible sensor is also subjected to the strain from 0% to 120% and then returns to 0% again (Fig. 5c). Moreover, the sensing performance after repeated stretch/release cycles at 120% strain was tested to further evaluate the stability and durability of the

sensor in practical applications. Figure 5d demonstrates that the $\Delta R/R_O$ value exhibits a stable response within a loading-unloading cycle of 2500 s, reflecting its excellent cyclic durability. To meet the demand and development of wearable flexible EMI sensors, monitoring and acquiring human motion signals is essential[6]. The sensor can be mounted on body areas such as the finger, wrist, knee, and neck to track human movements (Fig. 5e–h). During cyclic bending, the resistance change shows a consistent sensing pattern at each bending angle, and the maximum change of resistance increases with increasing joint bending angle. The resistance variation of each angle possesses repeatability, reliability, and good reversibility in the process of cycling and straightening. Notably, the pulses are detected through the sensor linked to the carotid artery (Fig. 5i), meaning that the coatings may provide good feedback for tiny motions. Therefore, the above stable sensing signals can foresee the prepared thin coatings to have better development in the broader field of human motion monitoring.

## Joule heating performance

Thermal conversion capabilities have emerged as an indispensable attribute for EMI shielding materials when applied in challenging environments such as aviation, aerospace, electricity infrastructure in alpine regions, and wearable heaters[50]. Therefore, the Joule heating performance of the MLM-S coating applied to a board was comprehensively examined.

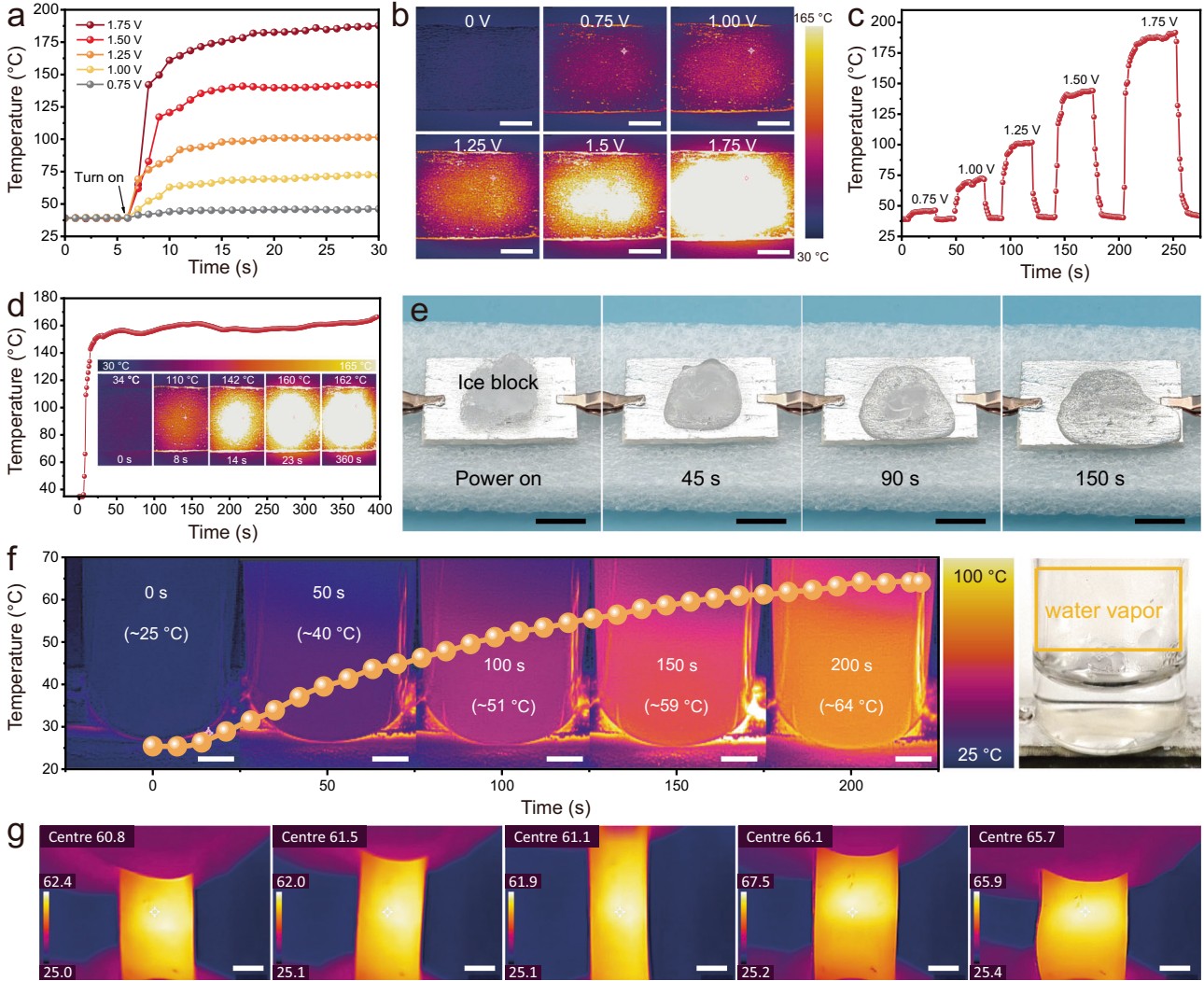

**Fig. 6 | Joule heating performance of the MXene/liquid metals composite prepared by solvent-assisted dispersion (MLM-S) on a board. a** Surface temperature change of MLM-S heater with distinct voltages. **b** Infrared (IR) images of the surface temperature of the MLM-S coating heater with distinct voltages. Scale bar: 5 mm. **c** The surface temperature of the heater is changed continuously on and off at different voltages. **d** Temperature stability of MLM-S coating under a driving voltage of 1.5 V standing. **e** Joule heating melts the surface ice. Scale bar: 10 mm. **f** IR and digital images of water heating processes. Scale bar: 5 mm. **g** IR images of a PDMS film coated with MLM-S under Joule heating at 0.85 V in the stretched state, a low voltage was chosen to avoid burning fingers. Scale bar: 5 mm.

Joule heating properties depend on the conductivity of the material[51]. The as-obtained MLM-S composites exhibit stable conductivity and high heating controllability (Supplementary Fig. 25 and Supplementary Fig. 26). Figure 6a exhibits the time-varying temperature profiles of the MLM-S coating under distinct direct current (DC) voltages. The surface saturation temperatures rapidly (within 10 s) reach 45, 70, 100, 140, and 175 °C and stabilize at the saturation level under 0.75, 1.00, 1.25, 1.50, and 1.75 V driving voltages, respectively. The infrared thermal images visually depict the saturation temperature of the MLM-S coating under varying driving voltages (Fig. 6b). This performance is mainly due to the excellent thermal conductivity of MXene and LM, as well as the perfect thermal conduction channel constructed by MLM-S[52]. When the voltage is further increased to 2.5 V, the center of the coating temperature can reach 300 ~ 400 °C and can even burn through the wood substrate (Supplementary Fig. 27). As exemplified in Fig. 6c, the surface temperature demonstrates regular cycles of ascent and descent during the loading and unloading of applied voltage, thereby showcasing the cycling stability. Furthermore, the temperature can be swiftly and conveniently adjusted by modulating the DC voltage in real-time (Supplementary Fig. 28), indicating the ability to respond quickly to the change of continuous voltages. The reliability of Joule

heat generation was further substantiated through a 6-minute stability test at a driving voltage of 1.5 V (Fig. 6d). The low driving voltage of the MLM-S coating ensures the safety of the human body, and the sensitive response guarantees less heating time. Simultaneously, the exceptional electric/thermal heating performance of MLM-S renders it suitable not only for maintaining warmth but also for anti-icing/deicing and water heating. An ice block was placed on the MLM-S coating to validate its practical viability, as shown in Fig. 6e. After 1.75 V voltage was applied, the ice block melted fast and completely melted after 150 s, while it took 15 min for the same volume of ice block to melt entirely at room temperature. Moreover, normal-temperature water was stored in a glass bottle and positioned on the coating heater depicted in Fig. 6f with a driving voltage of 1.5 V. It is evident that the temperature of the bottom water increased from the original 25–64 °C within 200 s, and the presence of water vapor is observed in the glass bottle. The heating process can be maintained continuously even when the MLM-S heater undergoes cyclic deformation (Fig. 6g, Supplementary Movie 8), which greatly guarantees the adaptability and reliability of the heater under the requirements of large deformation or cyclic deformation of the working environment. Moreover, the EMI shielding stability of the MLM-S film during the cyclic Joule heating process is important. As

shown in Supplementary Fig. 29, the EMI SE of the MLM-S film hardly decays after several cycles of power-on and power-off (1.5 V), indicating its high reliability. The combination of the low driving voltage, rapid thermal response, adaptability of complex surfaces, and long-term stability of MLM-S coating thus facilitates efficient thermal management in practical applications.

## Discussion
In summary, we proposed a convenient approach of incorporating MXene into LM through the utilization of SAD, which is distinguished from the conventional dispersion mechanisms employed in LM particle-based composite materials, as it achieves a more efficient and homogeneous dispersion of fillers through a solvent-provided slip layer and two-phase mutual dispersive wrapping. The unique malleability and printability of the prepared MLM conductive composite make it suitable for a variety of processing methods, demonstrating its ability to compete with traditional metal-based materials in printed electronics, outperforming polymer-based conductive materials in flexible stretchable devices, and exhibiting a high EMI shielding effectiveness. Specifically, the synthesized bi-conductive network MLM-S coating achieves 72 dB at only 5 μm and superb broadband shielding with an average of 105 dB at 20 μm, which can be attributed to the homogeneous conductive network of the MXene inside the coating and the synergistic dissipation of the high-conducting LM. Moreover, the MLM coating has satisfactory sensing and Joule heating performance, with a broad application prospect in flexible electronics and anti-icing/deicing. MLM composites can easily cope with complex structures and enable high design freedom, showing great potential in the field of advanced manufacturing for flexible functional electronics.

## Methods
### Chemicals and materials
EGaIn (LM, containing 75% Ga and 25% In, ≥ 99.99%) was purchased from Shenyang Jiabei Trading Co. Ltd. Ag nanosheets (>99.9%) were purchased from CMT New Materials (China Metallurgical Group). Lithium fluoride (LiF), Triiron Tetraoxide, (99.5%, 20 nm), and anhydrous ethanol (99.7%) were purchased from Shanghai Aladdin Industrial Corporation (China). Hydrochloric acid (HCl, 36–38%) was purchased from Sinopharm Chemical Reagents Co., Ltd. MAX powder ($Ti_3AlC_2$, 98%) was obtained from Foshan Xinxi Technology Co., Ltd. Multi-walled carbon nanotubes (å 97%, tube diameter: 3–15 μm, tube length: 15–30 μm) was purchased from Kaina Carbon New Material Co. Graphene (Single-story rate >80%, D90: 11–15 μm) was purchased from Shenzhen Guosen Pilot Technology Co. All chemicals without purity identification were of analytical grade without further purification.

### Fabrication of Ti3C2Tx (MXene)
The typical LiF-HCl etching method was used to fabricate multilayer $Ti_3C_2T_x$[53]. Firstly, 1.0 g of LiF was precisely weighed and then slowly added into 20 mL of HCl solution (12 M) in a Teflon hydrothermal reactor by magnetic stirring for 15 minutes. Secondly, 1.0 g of $Ti_3AlC_2$ powder was gradually added to the mixed solution and continuously stirred at 50 °C for 24 h. Next, remove the residual LiF in the mixed solution by several centrifugations (1325 g, 2 min per wash) until the pH of the supernatant was about 6. Then, the multilayer $Ti_3C_2T_x$ was sonicated in DI water for 1 h to obtain the few-layered $Ti_3C_2T_x$, followed by centrifuging at 745 g for half an hour to remove the residual multilayer $Ti_3C_2T_x$. Finally, the dark green supernatant containing $Ti_3C_2T_x$ nanosheets (16 mg mL$^{-1}$) was gathered and then stored in a refrigerator for further application.

### Synthesis of MXene/LM composites
**Dry powder mixing (DPM): MLM-D.** The prepared MXene dispersion was frozen at −20 °C and freeze drying at −80 °C for 72 h while the pressure was kept below 2 Pa. The obtained MXene aerogel was re-dispersed in anhydrous ethanol (EtOH) and broken by sonication using a cell crusher under nitrogen protection with a sonication power of 800 W and a sonication time of 30 min. Subsequent centrifugation was performed to obtain the precipitate, which was dried under vacuum for 24 h and ground into powder. The obtained MXene powder was gradually added to LM (ambient temperature) under magnetic stirring (1000 rpm) in a Teflon beaker until a homogeneous plasticine-like consistency was formed that no longer flowed, and the stirring could be stopped. The contents of MXene powder were 2, 4, 6, 8, 10 and 20 wt%. When the powder addition reached 20 wt%, the viscosity of the mixture was too high to stir and coated on paper. Meanwhile, Silver nanosheets were mixed with LM using the above method to form plasticine as a control group.

### Solvent-assisted dispersion (SAD): MLM-S
Distinguished from the traditional DPM, SAD does not require freeze drying and grinding of the MXene dispersion. The MXene nanosheets were flocculated by adding excess EtOH to the corresponding amount of MXene dispersion in each group, followed by centrifugation (5301 g, 5 min) to obtain MXene paste. The above procedure was repeated 3 times with the addition of EtOH to ensure that most of the deionized water was displaced. Finally, LM was added to the resulting MXene paste, heated and magnetically stirred (1000 rpm) for 9 h to evaporate EtOH, resulting in a homogeneous plasticine-like mixture completely. According to the mass ratio of LM accounted for by MXene ($x$ wt%), the obtained MLM-D and MLM-S plasticine can be labeled as $M_xLM$-D and $M_xLM$-S, respectively. The relevant data for LM and MXene content and EMI SE data of $M_xLM$-D are listed in Supplementary Table 3.

### Essential ethical statements
The individual involved in the sensing test was exclusively limited to the author himself (Haojie Jiang). The study about motion detection was approved by the Ethics Committee of East Hospital Affiliated with Tongji University (reference number 2016034). All included patients gave their oral and written informed consent.

## Data availability
Source data are provided with this paper.

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

## Acknowledgements

This work was supported by the National Key Research and Development Program of China (Grant: No. 2024YFE0100600, W.L.), the National Natural Science Foundation of China (No. 52373303, W.L.), the Shanghai Municipal Science and Technology Major Project (2021SHZDZX0100, W.L.), the Fundamental Research Funds for the Central Universities, and the Interdisciplinary Joint Research and Development Project of Tongji University (No. 2022-4-ZD-01, W.L.). Thanks to Ms. Diyue Zhang and Engineer Chunjie Cao for their support and help.

## Author contributions

W.L. guided and supervised the whole research. H.J. conceived the original idea; H.J., H.G., F.M., Y.W., and Z.X. performed the experiments and characterizations. F.P. and X.W. performed the microcosmic and compositional characterization. M.Y. and Y.G. performed the DFT calculation. L.R., S.Z., and Y.Y. analyzed the experimental data. Z.X., L.L., B.Y., and C.W. provided valuable guidance. H.J. and B.Y. wrote the manuscript. All authors participated in the discussion and interpretation of the data.

## Competing interests

The authors declare no competing interests.
