## [Peer Review File · Nature Communications]

Malleable, Printable, Bondable, and Highly Conductive MXene/Liquid Metal Plasticine with Improved WettabilityREVIEWER COMMENTS

Reviewer #1 (Remarks to the Author):

The manuscript entitled "Phagocytosis-inspired malleable, printable, bondable, and highly conductive MXene/Liquid Metal Plasticine with wide wettability" (NCOMMS-23-64024) by Haojie Jiang et al. has been reviewed. In this work, the authors proposed a novel method of incorporating MXene into LM (MLM) through the utilization of solvent-assisted dispersion (SAD). The obtained MLM coating has satisfactory sensing and Joule heating performance, with a broad application prospect in flexible electronics, sensor and anti-icing/deicing. However, major drawbacks still remain with the manuscript. Therefore, I recommend to further evaluation after modification. Some advices for improving the manuscript are as follows:

1. The author claims that the obtained MLM coating has the microstructure of the MXene LM intermolecular network (bulk heterojunction) shown in Figure 1, but although the author has made many characterizations, there is a lack of direct evidence. Due to the easy oxidation of LMs, the author did not describe how the oxide particles (Ga₂O₃ insulating oxide shell) are dispersed in the microstructure?.
2. The changes in contact angle during the mixing process of MXene and LM should be provided.
3. AFM images of NLM coatings, especially phase images and conductive atomic force images should be provided for further analysis of the composition and distribution of Mxene, LM, Ga₂O₃ in the coating.
4. Moreover, there is lack of in-depth analysis on the rheological evolution process.
5. There should be a space between scientific units and numerical values, but this convention is not adhered to in page 25, line 521.
6. Why the fitting peak in the high-resolution XPS spectra for the MLM and MXene in Fig 3a has a significant shift? The various peaks under the spectra represent various moieties assumed to exist. Please refer to the following literature: Applied Surface Science 362 (2016) 406-417; Electrochimica Acta 444 (2023) 142022.
7. After joule heating test, what will happen on the EMI shielding performance?
8. It is suggested to cite some of the relevant literature on liquid metal related joule heating application. Such as Adv. Mater. Interfaces, 2022, 2102266; Composites Communications 2023, 38, 101476.

Reviewer #2 (Remarks to the Author):

In this manuscript, the authors proposed a solvent-assisted dispersion method to integrate MXene into LM. The MXene/Liquid Metal Plasticine exhibits good electromagnetic interference shielding performance at 20 μm, facilitates direct printing. There are still several issues need to be addressed.

1. Page 22, line 462, the authors claimed that "we firstly proposed a novel approach of incorporating MXene into LM through the utilization of SAD, which is distinguished from the conventional dispersion mechanisms employed in LM particle-based composite materials". As a matter of fact, in a paper published in ACS Nano (Ref.17, 2022, 16, 14490-14502, doi:10.1021/acsnano.2c04863), the authors used ethyl acetate to fabricate MXene/magnetic liquid metal slurry.
2. On page 2, line 37-38, the authors claimed that " The existing LM-based functional materials are commonly assembled in dispersed core shell structures 17-19..." Actually, the materials mentioned in Ref. 18 (MXene-encapsulated magnetic liquid metal) is core-shell structure, while the materials developed in Ref. 17 is slurry uniformly dispersed with MXene and magnetic LM droplets, and Ref. 19 is a review paper. The rigor of expression issues needs to be strengthened.
3. The concept of phagocytosis is puzzling, according to the manuscript on page 6, the author's descriptions contradict each other. For example,

-Line 114, "the MXene paste (phagocyte)"

Line 125-127, "the fragmented and refined LM begins to gradually aggregate and come into contact

(forming a new phagocyte)"

It is found that the MXene paste is defined as phagocyte at the beginning, while aggregated LM is also defined as phagocyte. The definition of phagocyte appears to be unclear.

-Line 116-117, "the appropriate amount of LM (the target substance)"

Line 127-128, "which then engulfs the original MXene paste (the new target substance)."

It is really strange that the target substance is LM first, then changed to MXene paste.

In addition, the author described "mechanical agitation process" as lysosomes, it's hard to imagine describing a process as a functional part of a cell. The bionic mechanism of the material is confusing.

The authors should provide a clear description of the bionic mechanism of the material.

4. Figure 4i provides a comparison of SSEt between M2LM-S coating and other EMI shielding materials published, the data range for SSEt in the submitted work is reasonable, with magnitudes of 10^4 - 10^5 , however, the authors chose smaller SSEt value from the literature to emphasize the advantage of their results. A review paper once made a comprehensive summary figure displayed as SSE/t of MXene-based different structures as a function of thickness (Adv. Funct. Mater. 2020, 2000883, Figure 20c), which showed several MXene materials also have good SSEt performance. For example, Sr.# 26 listed in Table 1 reported a MXene structure with a thickness 0,018mm (thickness is 0.020 mm in this manuscript) has SSEt of 69444 (original Ref. is Adv. Mater. 2017, 29, 1702367), which is even higher than the data reported in this work.

Reviewer #3 (Remarks to the Author):

In this work, the authors reported a solvent-assisted dispersion (SAD) method to fabricate malleable material with good dispersion. The rheological properties of the MXene/LM (MLM) facilitated direct printing and adaptation to various structures. The MLM coating showed excellent electromagnetic interference (EMI) shielding properties (105 dB at 20 μ m) and joule heating performance. However, the innovation of this work is insufficient, since some similar work has reported related results (Science Advances 2021, 7, 3767; ACS Nano 2023, 17, 13, 12616, ACS Nano 2022, 16, 9, 14490.). It also lacks of some critical proof experiments. Thus, this manuscript can not meet the high requirement of Nature Communications. The following revisions need to be improved.

1. In the Introduction, the authors emphasize the importance of rheological modification of LM. It is suggested that the authors quantify the rheological properties of MLM-S.
2. The homogeneous dispersion of MXene is important for MLM. However, there is no proof in the manuscript about the interfacial stability of MXene and LM from a thermodynamic point of view. Please indicate the specific types of interactions that stabilize MXene and LM.
3. There is mentioned that "The attenuation of EWMs within the inner structure of the MLM-S coating can be attributed to various factors involving multiple reflections and scattering of the penetrating waves amidst the layers of MXene sheets, LM, and numerous heterogeneous interfaces." However, the authors have stated in the EM shielding equation section that SEM is negligible. The authors are expected to check whether there are multiple reflections within the material when the wavelength of the electromagnetic wave is much larger than the thickness of the material.
4. The biggest advantage of LM composites is good deformability and cyclic stability. It would be more beneficial for practical applications of MLM if the authors provide the serviceability of MLM films under cyclic deformation.
5. The manuscript mentions the solvent-assisted dispersion (SAD) method as a generalized method to prepare homogeneous LM composites. Please give the corresponding data to support it.

RESPONSE TO REVIEWERS' COMMENTS

Reviewer #1:

The manuscript entitled "Phagocytosis-inspired malleable, printable, bondable, and highly conductive MXene/Liquid Metal Plasticine with wide wettability" (NCOMMS-23-64024) by Haojie Jiang et al. has been reviewed. In this work, the authors proposed a novel method of incorporating MXene into LM (MLM) through the utilization of solvent-assisted dispersion (SAD). The obtained MLM coating has satisfactory sensing and Joule heating performance, with a broad application prospect in flexible electronics, sensor, and anti-icing/deicing. However, major drawbacks still remain with the manuscript. Therefore, I recommend to further evaluation after modification.

Answer: We thank the reviewer for his/her insights and valuable suggestions. We have provided additional experimental data on the microscopic feature and rheological properties of the MXene/liquid metal plasticine, and the stability of the dispersions as well.

Q1: The author claims that the obtained MLM coating has the microstructure of the MXene LM intermolecular network (bulk heterojunction) shown in Figure 1, but although the author has made many characterizations, there is a lack of direct evidence. Due to the easy oxidation of LMs, the author did not describe how the oxide particles (Ga₂O₃ insulating oxide shell) are dispersed in the microstructure?

Answer: We thank the reviewer for pointing this out. To address this point, we have conducted a set of characterizations (including X-ray microscopy (XRM), backscattered electron mode of SEM, and AFM) to provide more information on the distribution of the components in the microstructure. Corresponding figures and text were modified, as shown below.

Fig. 1 | Fabrication of the MXene/LM composite. (a) Schematic illustration of the preparation process of SAD. **(b)** X-ray microscopy (XRM) image demonstrating the bi-conductive phase structure of MLM-S. Two distinct MXene (blue) and LM (orange) domains interpenetrated.

Page 7: To further confirm the distribution of the two phases at the microscopic level, we used X-ray microscopy (XRM) to perform a Micro-CT scan of MLM-S. The density difference between MXene and LM allows the biphasic MLM structure to be visualized through the correlated density distributions, as measured by x-ray absorption (**Movie S1 and S2**). **Fig. 1b** shows the distribution of the two phases within a cube with a side length of 722 μm. MXene (blue) adheres to each other in a nearly spherical morphology, uniformly distributes within the LM (orange) matrix and forms continuous conductive paths. After software reconstruction, it is calculated that MXene occupies 34 % to 42 % of MLM-S by volume with a weight percentage of 2 wt%.

Fig. 3 | Microscopic observation of MLM-S. (a) The optical photograph of a weighing paper coated with MLM-S. (b) SEM image of the MLM-S coating surface. (b₁₋₄) EDS mapping of the region of (b). (c) Enlarged SEM image of the orange box region in (b). (c₁₋₄) EDS mapping of the region of (c). Schematic diagram of (d) MLM-S Plasticine for (e) blade coating. (f) The 3D image of MLM-S plasticine after section and reconstruction by X-ray microscopy (XRM). (g) The horizontal section of (f) obtained by XRM. (h) Backscattered electron image of MLM-S coating surface. (i) AFM phase image and (j) AFM 3D height image of M₂LM-S coating. (k) SEM image of the coating under bending at 90°.

Page 13: As presented in Fig. S14e and related EDS analysis (Fig. S14e₁₋₄), obvious MXene agglomerates (marked by Ti elemental signal aggregation) and numerous defects (holes and cracks) can be observed on the surface of the MLM-D coating. Furthermore, the surfaces of MXene are almost devoid of Ga/In/O elemental signals (Fig. S14e₂₋₄ and Fig. S14f₂₋₄), implying poor interaction between the two phases, leading to subsequent phase separation. As a comparison, MXene is uniformly distributed in the MLM-S coating without any noticeable agglomeration (Fig. 3b₁). Meanwhile, the MLM-S surface

consists of a compact layer of Ga-In, as shown in **Fig. 3b2-3**. It is noted that minute black cracks and defects are present on the surface of the coating, which can be found in the magnified image in **Fig. 3c, and** that the cracks do not penetrate through the coating. **Fig. 3c1** suggests a slight exposure of MXene, but Ga-In and its oxide layer remain uniformly enveloped on the surface (**Fig. 3c2-4**), demonstrating the absence of phase separation. This stability results from the robust hydrogen bonds between the LM oxide layer and MXene in the SAD-synthesized MLM-S, allowing both phases to deform synergistically under the shear forces applied during coating. The 3D image (**Fig. 3e**) and its horizontal section (**Fig. 3g**) obtained by XRM further reveal the spatial distribution of MXene within the plasticine. The near-spherical MXene, which is uniformly dispersed in LM, is extruded into an irregular morphology along with the extended LM on the substrate and constitutes a continuous network (**Fig. 3h**), proving the existence of a bicontinuous structure in the MLM-S layer. As seen in the AFM phase image (**Fig. 3i**), the changes in the vibrational phase of the probe due to the different material phases present different colors in the phase diagram, showing the composition and distribution of the coating. The light-white MXene region is wrapped by the black metal oxide layer, and the brown LM almost covers the whole scanning area. The AFM height image of the coating demonstrates the undulating surface morphology of the MXene therein, in contrast to the comparatively smooth LM regions. (**Fig. 3j**).

Q2: The changes in contact angle during the mixing process of MXene and LM should be provided.

Answer: This is a good point. We recorded the temporal evolution of the contact angle of the MXene/LM mixture on a PTFE plate (Fig. S3). It can be found that the pure LM is ineffective in wetting the surface, whereas the final MXene/LM mixture demonstrates effective wetting.

Fig. S3 The contact angle and optical photographs of each stage during SAD.

Page 7: The optical images and contact angle of each stage, presented in **Fig. S3**, illustrate that the introduction of MXene into LM by SAD improves the non-wetting behavior of LM on PTFE plates and enables complete spreading, demonstrating the adequate wetting of MLM-S on non-reactive interfaces.

Q3: AFM images of NLM coatings, especially phase images and conductive atomic force images should be provided for further analysis of the composition and distribution of Mxene, LM, Ga₂O₃ in the coating.

Answer: We agree with the reviewer. The conductive atomic force images need to be tested in contact mode using a special probe, and several institutions we consulted refused to test LM-based materials (LM can cause contamination of the probe), so we can not test. The phase images and morphology of M₂LM-S coating were characterized by conventional AFM in tapping mode and we hope to answer your question.

Fig. 3 (i) AFM phase image and **(j)** AFM 3D height image of M₂LM-S coating.

Page 13-14: As seen in the AFM phase image (**Fig. 3i**), the changes in the vibrational phase of the probe due to the different material phases present different colors in the phase diagram, showing the composition and distribution of the coating. The light-white MXene region is wrapped by the black metal oxide layer, and the brown LM almost covers the whole scanning area. The AFM height image of the coating demonstrates the undulating surface morphology of the MXene therein, in contrast to the comparatively smooth LM regions. (**Fig. 3j**).

Q4: Moreover, there is lack of in-depth analysis on the rheological evolution process.

Answer: We thank the reviewer for this. We have characterized the evolution of the rheological properties and performed corresponding in-depth analyses. Changes in the revised manuscripts are given below:

Fig. S10 Viscosity versus shear rate plots for the MLM-S with different MXene content. Inserts: High-viscosity MLM-S for coating (circled in blue) and low-viscosity MLM-S for printing (circled in red).

Page 10: Notably, the high surface tension of pure LM limits its microelectronic printability⁴³. The introduction of MXene improves the rheological properties of the LM which plays a vital role in the realization of direct printing. As shown in **Fig. S10**, the viscosity of MLM-S gradually increases with MXene content (from 82174 Pa·s for M_{0.25}LM-S to 814672 Pa·s for M₃LM-S at a shear rate of 0.013 s⁻¹). The transformation from a nonviscous liquid dispersion to a plasticine-like state was distinguished by a significant increase in viscosity and the absence of flow unless physically agitated, as illustrated by a sample in a tilted beaker (**Fig. S11**, **Movie S4**). Meanwhile, the viscosity of samples all decreases rapidly with the shear rate, exhibiting the shear-thinning behavior, which allows for continuous extrusion of MLM-S. The low-viscosity M_{0.25}LM-S can be used directly as an ink for rapid printing of high-fidelity 2D patterns without the need to mix with other insulating flexible substrates⁴⁴ (**Movie S5**).

Q5: There should be a space between scientific units and numerical values, but this convention is not adhered to in page 25, line 521.

Answer: Thanks for pointing it out. We have corrected it and applied this rule to the whole manuscript in the revision.

Q6: Why the fitting peak in the high-resolution XPS spectra for the MLM and MXene in Fig 3a has a significant shift? The various peaks under the spectra represent various moieties assumed to exist. Please refer to the following literature: Applied Surface Science 362 (2016) 406-417; Electrochimica Acta 444

(2023) 142022.

Answer: Thank you for your insightful comment. We have carefully addressed this issue based on the references you provided. The revised content is as follows:

Page 9: In contrast to the pristine MXene HR-XPS Ti 2p spectrum, the MLM composite exhibits additional subpeaks of In and In₂O₃ in the Ti 2p spectrum due to the overlapping region with In 3d_{3/2} (**Fig. 2a**), and the same occurs in the In 3d high-resolution spectrum of MLM (**Fig. S6c**). Moreover, the MLM composite exhibits a shift in the Ti 2p peaks towards lower binding energy. This decrease can be primarily attributed to the synergetic effects of defect formation and interfacial interactions between LM and MXene.^{40,41} The shift to lower binding energies suggests an increase in the electron density around the Ti atoms, which is attributed to the hydrogen-bonding interaction between the metal oxide layer (**Fig. S6**) and the hydroxyl groups on the MXene surface⁴², thus facilitating the charge transfer.

Q7: After joule heating test, what will happen on the EMI shielding performance?

Answer: This is an excellent question. To answer it, we have conducted additional experiments. Specifically, we tracked the EMI shielding performance of a M₂LM-S coated paper during 100 joule heating cycles. The results show that the shielding performance stayed almost constant, indicating excellent thermal cycling stability.

Fig. S30 EMI SE of the M₂LM-S coating after 100 joule heating cycle.

Page 2: Moreover, the EMI shielding stability of the MLM-S film during the cyclic Joule heating process is important. As shown in **Fig. S30**, the EMI SE of the MLM-S film hardly decays after several cycles of power-on and power-off (1.5 V), indicating its high reliability.

Q8: It is suggested to cite some of the relevant literature on liquid metal related joule heating application. Such as Adv. Mater. Interfaces, 2022, 2102266; Composites Communications 2023, 38, 101476.

Answer: This is a very good suggestion. Relevant references were added in the section of Joule heating.

Page 23: Joule heating properties depend on the conductivity of the material.⁵¹ The as-obtained MLM-S composites exhibit stable conductivity and high heating controllability (**Fig. S26** and **Fig. S27**).

Page 23: This performance is mainly due to the excellent thermal conductivity of MXene and LM, as well as the perfect thermal conduction channel constructed by MLM-S.⁵²

Reviewer #2:

In this manuscript, the authors proposed a solvent-assisted dispersion method to integrate MXene into LM. The MXene/Liquid Metal Plasticine exhibits good electromagnetic interference shielding performance at 20 μm , facilitates direct printing. There are still several issues need to be addressed.

Answer: We thank the reviewer for his/her constructive feedback, which we have taken very seriously during the revision process. We hope the changes we made will be to the reviewer's satisfaction.

Q1: Page 22, line 462, the authors claimed that “we firstly proposed a novel approach of incorporating MXene into LM through the utilization of SAD, which is distinguished from the conventional dispersion mechanisms employed in LM particle-based composite materials”. As a matter of fact, in a paper published in ACS Nano (Ref.17, 2022, 16, 14490-14502, doi:10.1021/acsnano.2c04863), the authors used ethyl acetate to fabricate MXene/magnetic liquid metal slurry.

Answer: We sincerely appreciate the comment. In the reference mentioned above (ACS Nano 2022, 16, 9, 14490–14502), the researchers prepared a three-phase composite based on the polymer Poly(styrene-butadiene-styrene) (SBS), in which ethyl acetate was used to dissolve the SBS so that the MXene powder and the LM could be dispersed in the SBS and the elastic fibers were prepared by extrusion through a syringe. Thus, the above material is suitable for preparing shielding fabric. However, the three-phase composites cannot be molded without SBS. In addition, LM is somewhat isolated from each other by the insulating polymer matrix (SBS), thus large amount of LM is needed to establish an effective conductive network (i.e., the mass of LM needs to be several times that of the SBS).

The composites in the above literature are fundamentally different from the MXene/LM composites in our paper in terms of the material molding process and dispersion mechanism. In our work, LM is used as the matrix to disperse MXene, wherein MXene is not dispersed in the form of traditional powder, which avoids the problem that the powder is not easily dispersed uniformly in the LM matrix after agglomeration. The resulting MLM composites are malleable, printable, bondable and show excellent wettability on a variety of substrate surfaces.

Therefore, our work is distinctly different from the work mentioned above from the perspective of the underlying dispersion mechanism, the role of individual components, the composition and properties, and suitable shaping/patterning methods.

Q2: On page 2, line 37-38, the authors claimed that “ The existing LM-based functional materials are commonly assembled in dispersed core shell structures 17-19...” Actually, the materials mentioned in Ref. 18 (MXene-encapsulated magnetic liquid metal) is core-shell structure, while the materials developed in Ref. 17 is slurry uniformly dispersed with MXene and magnetic LM droplets, and Ref. 19 is a review paper. The rigor of expression issues needs to be strengthened.

Answer: We thank the reviewer for pointing this out. We have revised the relevant content with scientific

rigor. The revised contents are as follows:

Page 2: Existing LM-based conductive functional materials typically involve interface engineering of LM droplets followed by blending with reinforcing phases¹⁷⁻²² or adding LM composites²³⁻²⁵ to insulating substrates in order to achieve specific shapes and mechanical strength.

Page 2: Although the LM conduction path can be constructed directly in the matrix^{10,16,31}, it increases the complexity and technical difficulty of the preparation process.

The material systems of the literatures mentioned above are as follows:

Ref. 17: Graphene oxide sheets encapsulating LM droplets, then coated by PDMS to be a film.

Ref. 18: LM droplets are coated by MWCNTs and dispersed in a polyacrylic elastomer.

Ref. 19: MXene encapsulates LM droplets and then dispersed in a polyacrylamide matrix.

Ref. 20: LM droplets are encapsulated by the carbon layer, and then coated by PDMS.

Ref. 21: LM is stirred in SBS dispersion together with MXene powder, then extruded and cured into fiber.

Ref. 22: GO encapsulated LM-based dual network hydrogel.

Ref. 23: LM and Fe powder are mixed together and then added to PDMS to make elastomers.

Ref. 24: LM and Fe powder are mixed together and then added to Ecoflex to make elastomers.

Ref. 25: LM and MXene/Ag nanowires were sprayed on the pre-stretched PDMS matrix by spray gun.

Ref. 10: The wetting-induced assembly method based on the differential capillary effect for liquid metal ink is created to fill the PDMS micropore channels.

Ref. 16: A coaxial printer is used to print liquid metal coated with PDMS.

Ref. 31: The LM and PDMS were used for dual-material synchronous 3D printing to construct an elastomer.

Q3: The concept of phagocytosis is puzzling, according to the manuscript on page 6, the author's descriptions contradict each other. For example,

-Line 114, “the MXene paste (phagocyte)”

Line 125-127, “the fragmented and refined LM begins to gradually aggregate and come into contact (forming a new phagocyte)”

It is found that the MXene paste is defined as phagocyte at the beginning, while aggregated LM is also defined as phagocyte. The definition of phagocyte appears to be unclear.

-Line 116-117, “the appropriate amount of LM (the target substance)”

Line 127-128, “which then engulfs the original MXene paste (the new target substance).”

It is really strange that the target substance is LM first, then changed to MXene paste.

In addition, the author described “mechanical agitation process” as lysosomes, it's hard to imagine describing a process as a functional part of a cell. The bionic mechanism of the material is confusing.

The authors should provide a clear description of the bionic mechanism of the material.

Answer: We sincerely appreciate the reviewer’s comment on the formation mechanism of MLM materials and we apologize for any confusion caused by the analogy used in the article. The editor also suggested deleting the bionic section. We agree with the reviewer and the editor that the inclusion of the phagocytosis may obscure the strengths of our work and we understand it is of great importance to focus on the core contributions of our work. Therefore, we have removed all contents about the biomimetic section in the revised manuscript.

To clarify the biomimetic mechanism and address the reviewer’s concerns, we annotated the schematics of the MLM-S formation process. The dispersion process can be divided into two parts (Part 1 and Part 2), so the phagocytes used for analogy are not unique and will be transformed automatically during the dispersion process. Before the EtOH is completely evaporated (Part 1), the MXene paste occupies a large volume, and the LM is gradually dispersed in the MXene paste with the stirring process, so at this stage, the MXene paste is the phagocyte, and LM is the phagocytosed material. After the gradual volatilization of EtOH (Part 2), the volume of MXene paste decreases dramatically. The fragmented and refined LM begins to contact and aggregate to occupy a larger volume and inversely wraps the MXene. So, it can be assumed that the LM is the phagocytosis cell at this stage, and the MXene is the phagocytosed material. Moreover, as for the analogy to the lysosome, our original intention was to liken the process of lysosomal degradation of bacteria to the process of dispersing the target substance through mechanical stirring, but this analogy caused confusion, for which we apologize again.

Part 1

EtOH evaporation

Part 2

Q4: Figure 4i provides a comparison of SSEt between M2LM-S coating and other EMI shielding materials published, the data range for SSEt in the submitted work is reasonable, with magnitudes of 10^4 - 10^5 , however, the authors chose smaller SSEt value from the literature to emphasize the advantage of their results. A review paper once made a comprehensive summary figure displayed as SSEt of MXene-based different structures as a function of thickness (Adv. Funct. Mater. 2020, 2000883, Figure 20c), which showed several MXene materials also have good SSEt performance. For example, Sr.# 26 listed in Table 1 reported a MXene structure with a thickness 0,018mm (thickness is 0.020 mm in this manuscript) has SSEt of 69444 (original Ref. is Adv. Mater. 2017, 29, 1702367), which is even higher than the data reported in this work.

Answer: We are thankful to the reviewer for pointing this out. We have extensively searched relevant references in the literature and expanded the number of references used for the comparison in hope that a fair and comprehensive comparison can be achieved. Specifically, we incorporated the SSE/t of MXene structures reported in *Adv. Funct. Mater.* 2020, 2000883 and *Adv. Mater.* 2017, 29, 1702367. We also discussed the cause of the performance difference, such as the difference between M2LM-S coating and other MXene-based materials. The related modifications are given below:

Page 19: Dense MXene films and air-introduced MXene aerogels/foams provide extremely excellent shielding performance, but further improvements are needed in mechanical strength, flexibility, and durability. The composite of MXene and the reinforcing phase improved the overall strength, but the bucket effect needs to be better addressed.

Fig. 4(i) Comparison of SSE_t between M2LM-S coating and other EMI shielding materials published in the literature.

Fig. S23 Comparison of EMI SE between M₂LM-S coating and other EMI shielding materials published in the literature.

Table S2 Comparison of EMI shielding performance for various materials.

Materials	Thickness (μm)	EMI SE in X-band [dB]	SSE _t [dB cm ² g ⁻¹]	Ref.
Cu foil	3100	90	32	12
Stainless steel foil	4000	89	28	12
Ag foil	10	58	5576	13
Al foil	50	63	4630	13
Graphene	50	60	6000	14
Graphene(annealed)	60	90	7500	15

Flexible graphite	790	102	645.6	16
	3100	130	209.7	16
MXene/CNF	35	40	7029	17
MXene/AgNWs/ANF	45	48.1	10688.9	18
MXene/BC	9	37.7	21429	19
MXene/NR	54	63.5	18989.8	20
MXene/CNF/MWCNT	43	45.1	19543.17	21
MXene/PVA	27	44.4	9343	22
MXene/PT	30	32.0	4085.92	23
Ti ₃ C ₂ T _x film	40	83.5	9488.64	24
Heat-treated Ti ₃ C ₂ T _x film	40	93.0	10568.2	24
Ti ₃ CNT _x film	40	61.4	6977.3	24
Heat-treated Ti ₃ CNT _x film	40	116.2	13204.6	24
Ti ₃ C ₂ T _x aerogel	1000	70.6	64182	25
Ti ₃ CNT _x aerogel	1000	69.2	62909	25
Ti ₂ CT _x aerogel	1000	54.1	49182	25
Ti ₃ C ₂ T _x foam	6	32	136752	26
Ti ₃ C ₂ T _x foam	18	50	69444	26
Ti ₃ C ₂ T _x foam	6	70	53030	26
Mxene/SA/Ca ²⁺	2.8	46	58929	27
LM/CNT fabrics	20	68	8500	28
LM/CNT coating	5	50.9	22622.2	29

LM/rGO coating	20	66	24812	6
LM/rGO film	67	100	7462.7	30
LM/MXene textile	430	47.1	1095.3	31
LM/CNTs/WPU	200	70	5984.4	32
	3	68	59336.8	
	5	76	39790.6	This
LM/MXene plasticine	10	85	22251.3	work
	20	109	14267.1	

Reviewer #3:

In this work, the authors reported a solvent-assisted dispersion (SAD) method to fabricate malleable material with good dispersion. The rheological properties of the MXene/LM (MLM) facilitated direct printing and adaptation to various structures. The MLM coating showed excellent electromagnetic interference (EMI) shielding properties (105 dB at 20 μm) and joule heating performance. However, the innovation of this work is insufficient, since some similar work has reported related results (Science Advances 2021, 7, 3767; ACS Nano 2023, 17, 13, 12616, ACS Nano 2022, 16, 9, 14490.). It also lacks of some critical proof experiments. Thus, this manuscript can not meet the high requirement of Nature Communications. The following revisions need to be improved.

Answer: We thank the reviewer for such a thorough review of our work and the constructive comments that have helped improve the quality of our paper significantly.

First of all, we would like to elaborate on the distinctions and innovations of this paper in comparison with the three literatures you mentioned:

- 1、 Science Advances 2021, 7,3767: This article reports a method to make liquid metal composites by vigorously mixing gallium (Ga) with non-metallic particles of graphene oxide, graphite, diamond, and silicon carbide. The resulting liquid metal composites display either a paste or putty-like behavior depending on the composition. The approach in this work deviates from the traditional ones of exclusively employing composite metal particles. Instead, it introduces a novel route to compound a variety of non-metallic materials to form a putty-like substance. However, it is mentioned in its Introduction section that carbon nanotubes and MXene cannot be directly mixed with LM unless special surface modifications are made. The authors of the paper attempted to mix carbon nanotubes with LM but were unsuccessful (shown in the supporting information of the paper).
- 2、 ACS Nano 2023, 17, 13, 12616-12628: In this article, a GO-bridged LM was prepared and formed into a conductive film through subsequent sintering under high-temperature and high-pressure. The film was then coated with PDMS. The resulting composite film has excellent multifunctionality. However, it should be noted that the preparation process of the composite requires both a high-temperature and a high-pressure sintering process to enable GO reduction as well as liquid metal conduction, respectively. The pressure sintering process is essential for this approach regardless of whether the GO is replaced with some other substance. Therefore, we mentioned in the Introduction of our original manuscript: “Furthermore, the limited wettability of LM at non-reactive interfaces and the secondary sintering required for conductive path activation pose challenges in their shaping on flexible substrates or complex surfaces.”. Additionally, the size of the functional film producible via this method is limited by the pressure sintering equipment, meaning preparing a film with a large area using this approach is inherently challenging.
- 3、 ACS Nano 2022, 16,9,14490: This article reports a method to prepare a three-phase composite based

on poly(styrene-butadiene-styrene) (SBS), in which ethyl acetate was used to dissolve the SBS to facilitate the dispersion of MXene powder and LM. Elastic fibers can be prepared by extrusion of the composite through a syringe. The prepared elastic fiber has a good application prospect in fabricating multifunctional fabric. However, the three-phase composites cannot be molded in the absence of SBS, and the LM is somewhat isolated from each other by the insulating polymer matrix (SBS), which necessitates the addition of LM at an amount that is several times that of SBS by mass in order to create an effective conductive network. The composites are fundamentally different from the MXene/LM composites in our study in terms of dispersion mechanism and composite molding method. In our work, LM is used as the matrix to disperse MXene. During the composite preparation process, MXene is not dispersed in the traditional powdered form, but rather in a dispersed wet state, which bypasses the challenge that the MXene powder cannot be dispersed uniformly in the LM matrix after agglomeration.

To sum up, our work addresses the key challenges faced by the methods reported in the three references above: a wide variety of functional materials cannot be directly composited with LM without surface modification (reference #1); cumbersome secondary post-processing is required to establish a conductive network in the composites and it poses a limitation on the scalability of fabrication and patterning flexibility of the composite (reference #2); the high proportion of LM in the composite is required to produce a conductive network in the composite and the composite contains a non-conductive polymeric substance as an additive (reference #3). Moreover, the MLM composites in our work have excellent malleability, bondability, wettability, and more importantly, printability.

Finally, we have added corresponding characterization data in response to the lack of direct evidence raised by the reviewer. We sincerely hope that our revised manuscript will meet the high requirements of Nature Communications. We thank the reviewer once again for his/her valuable feedback, which has spurred us to significantly improve our paper.

Q1: In the Introduction, the authors emphasize the importance of rheological modification of LM. It is suggested that the authors quantify the rheological properties of MLM-S.

Answer: We agree with the reviewer. This point was also raised by reviewer #1, Q4. We kindly refer the reviewer to the responses and modifications we made above.

Q2: The homogeneous dispersion of MXene is important for MLM. However, there is no proof in the manuscript about the interfacial stability of MXene and LM from a thermodynamic point of view. Please indicate the specific types of interactions that stabilize MXene and LM.

Answer: We thank the reviewer for his/her insightful comment. To investigate the possible interactions between the MXene and LM, we performed both experimental (XPS and FTIR) and computational (molecular dynamics simulation) studies. On top of the van der Waals interaction that universally presents in all the systems, hydrogen bonding between the MXene and LM was identified by our studies.

Given the intimate contact between the MXene and LM (as shown in Figure 3), we infer that the van der Waals interaction between them likely plays a significant role in their overall interaction. We include here the modifications we made for convenience.

Fig. 2 (a) Comparison of Ti 2p high-resolution XPS spectra of MLM composites and MXene. Simulation of interface interaction between LM and MXene at (b) 0 ps and (c) 1.7 ps.

Fig. S7 FTIR spectra of LM, MXene, and MLM.

Fig. S8 The system energy variation with time in the AIMD process.

Page 9-10: On top of the van der Waals interaction that are prevalent in all the systems, the stable binding between MXene and LM is critical and determines whether MLM-S can stay stable. The synthetic MLM composite was detected by X-ray diffraction (XRD) to investigate the phase composition (**Fig. S5**), indicating no other new phases were formed during the formation of MLM plasticine. X-ray photoelectron spectroscopy (XPS) was carried out to determine the surface chemical states of pristine materials and synthetic plasticine. In contrast to the pristine MXene HR-XPS Ti 2p spectrum, the MLM composite exhibits additional subpeaks of In and In_2O_3 in the Ti 2p spectrum due to the overlapping region with In $3d_{3/2}$ (**Fig. 2a**), and the same occurs in the In 3d high-resolution spectrum of MLM (**Fig. S6c**). Moreover, the MLM composite exhibits a shift in the Ti 2p peaks towards lower binding energy. This decrease can be primarily attributed to the synergetic effects of defect formation and interfacial interactions between LM and MXene.^{40,41} The shift to lower binding energies suggests an increase in the electron density around the Ti atoms, which is attributed to the hydrogen-bonding interaction between the metal oxide layer (**Fig. S6**) and the hydroxyl groups on the MXene surface⁴², thus facilitating the charge transfer. The redshift and broadening of hydroxyl peaks in the MLM infrared spectrum also prove the presence of hydrogen bonds (**Fig. S7**).

To further elucidate the bonding between MXene and LM interfaces, DFT simulations were conducted using VASP to investigate the interfacial interactions between the hydroxyl-terminated MXene and LM with an oxide layer. Specifically, six oxygen atoms were introduced into an amorphous $\text{Ga}_{2.5}\text{In}$ alloy model to simulate the formation of an oxide layer. After a simulation period of 1.7 picoseconds (**Fig. 2b, c**), hydrogen bonds were observed between two oxygen atoms within the metal oxide layer and the

hydroxyl terminations of MXene, ensuring a strong integration of two phases. Notably, the system energy stabilized after 1.7 picoseconds in a 2.2-picosecond dynamic equilibrium process (Fig. S8), further confirming the stability of the interface between MXene and EGaIn.

Therefore, the homogeneous dispersion of MXene in LM and the stable combination of the two phases prevent MLM-S from phase segregation and phase precipitation. Moreover, MLM-S retains the outstanding self-healing properties of LM, exhibiting remarkable processability and versatility.

Q3: There is mentioned that “The attenuation of EWMs within the inner structure of the MLM-S coating can be attributed to various factors involving multiple reflections and scattering of the penetrating waves amidst the layers of MXene sheets, LM, and numerous heterogeneous interfaces.” However, the authors have stated in the EM shielding equation section that SEM is negligible. The authors are expected to check whether there are multiple reflections within the material when the wavelength of the electromagnetic wave is much larger than the thickness of the material.

Answer: This is an excellent point. In cases where the wavelength of the electromagnetic wave is much larger than the material's thickness, the contribution of multiple internal reflections to the attenuation of electromagnetic waves can be considered negligible. Based on this, we deleted the text involving multiple reflections and re-summarized the shielding mechanism as follows:

Page 18: The EMI shielding capability stems from the exceptional conductivity exhibited by its surface, resulting from the complete wrapping of LM. Therefore, most of the EMWs are reflected at the interface because of the impedance mismatch, which is closely related to the presence of numerous free charge carriers. Additionally, the presence of the LM/MXene bi-conductive structure further augments its shielding capabilities. The residual EMWs infiltrate the depths of the coating, where the interconnected conductive network shows a remarkable ability to dissipate EWM, inducing a significant induced current, thereby converting a substantial portion of the EWMs energy into thermal energy. Moreover, the accumulation of free charges at the interfaces between the MXene sheets and the LM induces interfacial polarization, further contributing to the attenuation process. Almost all of the penetrated EWMs are attenuated inside the structure, causing the T value close to 0.^{17,45,48}

Q4: The biggest advantage of LM composites is good deformability and cyclic stability. It would be more beneficial for practical applications of MLM if the authors provide the serviceability of MLM films under cyclic deformation.

Answer: We agree with the reviewer. In order to access the serviceability of MLM films under cyclic deformation, we conducted two additional sets of experiments: 1) the electromagnetic shielding performance of the MLM film as a function of ‘number of stretching cycles’ (up to 1000 cycles), and 2) Joule heating performance of MLM-S coatings under cyclic stretching and bending (The heater was deformed and stretched by hand, so a low voltage (0.85V) was chosen to avoid burns, Movie S8).

The findings from the first experiment demonstrate that MLM film can consistently uphold its excellent

shielding performance even after undergoing 1000 stretching cycles. The results from the second experiment show that there is no discernable performance degradation. These two experiments compellingly demonstrate the stability and durability of the MLM film.

Fig. S21 EMI SE of Ecoflex films coated with M₂LM-S after stretching 1000 cycles.

Page 18: From a practical application perspective, reliable EMI shielding is critical. Benefiting from the deformability and cyclic stability of the LM matrix, the shielding performance of Ecoflex films coated with M₂LM-S remains stable after undergoing 1000 cycles of 100 % stretching (**Fig. S21**).

Fig. 6 (g) IR images of a PDMS film coated with MLM-S under Joule heating at 0.85 V in the stretched state, a low voltage was chosen to avoid burning fingers.

Fig. S30 EMI SE of the M₂LM-S coating after 100 joule heating cycle.

Page 24: The heating process can be maintained continuously even when the MLM-S heater undergoes cyclic deformation (**Fig. 6g, Movie S8**), which greatly guarantees the adaptability and reliability of the heater under the requirements of large deformation or cyclic deformation of the working environment. Moreover, the EMI shielding stability of the MLM-S film during the cyclic Joule heating process is important. As shown in **Fig. S30**, the EMI SE of the MLM-S film hardly decays after several cycles of power-on and power-off (1.5 V), indicating its high reliability.

Q5: The manuscript mentions the solvent-assisted dispersion (SAD) method as a generalized method to prepare homogeneous LM composites. Please give the corresponding data to support it.

Answer: This is a wonderful point, which allows us to make our paper more impactful. The corresponding data is provided below to support the generality of our SAD method in preparing LM composites. Three representative nanoscale fillers, i.e., carbon nanotubes, graphene, and nano Fe₃O₄, were selected for the demonstration.

Page 6: We provide three successful cases of particle infiltration, which are difficult-to-be-dispersed multi-walled carbon nanotubes, graphene, and nano Fe₃O₄ particles (**Table S1**).

Table S1 SAD was used to inject the conventional unmixed fillers into LM

Filler	Mass loading	Result
Multi-walled carbon nanotubes	1.0 wt%	Graphene	1.0 wt%	Nono Fe ₃ O ₄	2.0 wt%	
REVIEWERS' COMMENTS

Reviewer #1 (Remarks to the Author):

The authors have clarified the reviewer's questions well and the manuscript can be published as it is.

Reviewer #2 (Remarks to the Author):

The authors have refined the manuscript further in response to the reviewers' comments. However, the work lacks innovation and is not suitable for publication in high-impact journals Nature Communications. Firstly, this article is akin to that of Science Advances 2021, 7, 3767 and ACS Nano 2022, 16, 9, 14490. Furthermore, the electromagnetic shielding performance of the new material is not markedly superior to previous works.

Reviewer #3 (Remarks to the Author):

This manuscript has been well revised and could be accepted.

RESPONSE TO REVIEWERS' COMMENTS

Response to the Reviewer #2

Reviewer #2:

The authors have refined the manuscript further in response to the reviewers' comments. However, the work lacks innovation and is not suitable for publication in high-impact journals Nature Communications. Firstly, this article is akin to that of Science Advances 2021, 7, 3767 and ACS Nano 2022, 16, 9, 14490. Furthermore, the electromagnetic shielding performance of the new material is not markedly superior to previous works.

Answer: We appreciate your concerns regarding the innovation of our work. We want to address these concerns by highlighting the key differences and novel contributions of our study compared to the cited works.

1. Fundamentally Different Methodology and Resulting Product:

Science Advances 2021, 7, 3767: This study focuses on the creation of liquid metal (LM) composites by mixing gallium with non-metallic particles like graphene oxide, graphite, diamond, and silicon carbide. The emphasis is on whether the target non-metallic particles and LM are miscible. However, it is mentioned in this article that CNTs and MXene cannot be directly mixed with LM unless special surface modifications are made. In addition, this study attempted but failed to composite CNTs with LM.

ACS Nano 2022, 16, 9, 14490: This work aims to enhance the conductivity of insulative elastic fibers by incorporating LM droplets and MXene. The focus is on modifying the elastic fiber substrates rather than dispersing low-dimensional fillers uniformly in LM. Therefore, it is unclear if and how well MXene integrates with LM in this study.

Our Study: We introduce a general solvent-assisted dispersion (SAD) method for stably and uniformly dispersing low-dimensional fillers, such as MXene and CNTs, into LM without the need for chemical modification. The result is a highly conductive (conductivity is similar to LM), malleable, and printable filler/LM plasticine with excellent wettability. Thus, both our methodology and the resulting product differ fundamentally from the two aforementioned articles.

2. Different Target Applications:

Science Advances 2021, 7, 3767: Primarily targets the thermal and electromagnetic applications of putty-like LM composites.

ACS Nano 2022, 16, 9, 14490: Focuses on wearable applications for electronic textiles.

Our Study: Our work provides a facile method for developing multifunctional flexible/stretchable devices. Specifically, the malleability and printability of the resulting product (MLM composites) greatly

simplify the device fabrication process, offering tremendous potential for advancing multifunctional flexible/stretchable devices.

In conclusion, we believe these differences underscore the innovation and originality of our research.

Additional Concerns on EMI Shielding Performance:

As shown in Fig. 4 of our manuscript, electromagnetic shielding performance of the LM composite prepared in this study notably competitive compared to state-of-the-art shielding materials. In particular, it surpasses the majority of them, and is among the highest under low thickness. More importantly, compared with low-density non-metallic functional materials, which are generally fragile and/or lack stretchability, our product strikes a fine balance between shielding performance, mechanical strength, flexibility, stretchability, and durability.